# PAC Reinforcement Learning with Rich Observations

**Akshay Krishnamurthy**
University of Massachusetts, Amherst
Amherst, MA, 01003
akshay@cs.umass.edu

**Alekh Agarwal**
Microsoft Research
New York, NY 10011
alekha@microsoft.com

**John Langford**
Microsoft Research
New York, NY 10011
jcl@microsoft.com

## Abstract

We propose and study a new model for reinforcement learning with rich observations, generalizing contextual bandits to sequential decision making. These models require an agent to take actions based on observations (features) with the goal of achieving long-term performance competitive with a large set of policies. To avoid barriers to sample-efficient learning associated with large observation spaces and general POMDPs, we focus on problems that can be summarized by a small number of hidden states and have long-term rewards that are predictable by a reactive function class. In this setting, we design and analyze a new reinforcement learning algorithm, Least Squares Value Elimination by Exploration. We prove that the algorithm learns near optimal behavior after a number of episodes that is polynomial in all relevant parameters, logarithmic in the number of policies, and independent of the size of the observation space. Our result provides theoretical justification for reinforcement learning with function approximation.

## 1 Introduction

The Atari Reinforcement Learning research program [21] has highlighted a critical deficiency of practical reinforcement learning algorithms in settings with rich observation spaces: they cannot effectively solve problems that require sophisticated exploration. How can we construct Reinforcement Learning (RL) algorithms which effectively plan and plan to explore?

In RL theory, this is a solved problem for Markov Decision Processes (MDPs) [6, 13, 26]. Why do these results not apply?

An easy response is, "because the hard games are not MDPs." This may be true for some of the hard games, but it is misleading—popular algorithms like $Q$-learning with $\epsilon$-greedy exploration do not even engage in minimal planning and global exploration[1] as is required to solve MDPs efficiently. MDP-optimized global exploration has also been avoided because of a polynomial dependence on the number of unique observations which is intractably large with observations from a visual sensor.

In contrast, supervised and contextual bandit learning algorithms have *no* dependence on the number of observations and at most a logarithmic dependence on the size of the underlying policy set. Approaches to RL with a weak dependence on these quantities exist [15] but suffer from an exponential dependence on the time horizon—with $K$ actions and a horizon of $H$, they require $\Omega(K^H)$ samples. Examples show that this dependence is necessary, although they typically require a large number of states. Can we find an RL algorithm with no dependence on the number of unique observations and a polynomial dependence on the number of actions $K$, the number of necessary states $M$, the horizon $H$, and the policy complexity $\log(|\Pi|)$?

To begin answering this question we consider a simplified setting with episodes of bounded length $H$ and deterministic state transitions. We further assume that we have a function class that contains the optimal observation-action value function $Q^\star$. These simplifications make the problem significantly more tractable without trivializing the core goal of designing a $\text{Poly}(K, M, H, \log(|\Pi|))$ algorithm. To this end, our contributions are:

1. A new class of models for studying reinforcement learning with rich observations. These models generalize both contextual bandits and small-state MDPs, but do not exhibit the partial observability issues of more complex models like POMDPs. We show *exponential lower bounds* on sample complexity in the absence of the assumptions to justify our model.

2. A new reinforcement learning algorithm Least Squares Value Elimination by Exploration (LSVEE) and a PAC guarantee that it finds a policy that is at most $\epsilon$ sub-optimal (with the above assumptions) using $\mathcal{O}\left(\frac{MK^2H^6}{\epsilon^3}\log(|\Pi|)\right)$ samples, with no dependence on the number of unique observations. This is done by combining ideas from contextual bandits with a novel state equality test and a global exploration technique. Like initial contextual bandit approaches [1], the algorithm is computationally inefficient since it requires enumeration of the policy class, an aspect we hope to address in future work.

LSVEE uses a function class to approximate future rewards, and thus lends theoretical backing for reinforcement learning with function approximation, which is the empirical state-of-the-art.

## 2  The Model

Our model is a **Contextual Decision Process**, a term we use broadly to refer to any sequential decision making task where an agent must make decision on the basis of rich features (context) to optimize long-term reward. In this section, we introduce the model, starting with basic notation. Let $H \in \mathbb{N}$ denote an episode length, $\mathcal{X} \subseteq \mathbb{R}^d$ an observation space, $\mathcal{A}$ a finite set of actions, and $\mathcal{S}$ a finite set of latent states. Let $K \triangleq |\mathcal{A}|$. We partition $\mathcal{S}$ into $H$ disjoint groups $\mathcal{S}_1, \ldots, \mathcal{S}_H$, each of size at most $M$. For a set $P$, $\Delta(P)$ denotes the set of distributions over $P$.

### 2.1  Basic Definitions

Our model is defined by the tuple $(\Gamma_1, \Gamma, D)$ where $\Gamma_1 \in \Delta(\mathcal{S}_1)$ denotes a starting state distribution, $\Gamma : (\mathcal{S} \times \mathcal{A}) \to \Delta(\mathcal{S})$ denotes the transition dynamics, and $D_s \in \Delta(\mathcal{X} \times [0,1]^K)$ associates a distribution over observation-reward pairs with each state $s \in \mathcal{S}$. We also use $D_s$ to denote the marginal distribution over observations (usage will be clear from context) and use $D_{s|x}$ for the conditional distribution over reward given the observation $x$ in state $s$. The marginal and conditional probabilities are referred to as $D_s(x)$ and $D_{s|x}(r)$.

We assume that the process is *layered* (also known as loop-free or acyclic) so that for any $s_h \in \mathcal{S}_h$ and action $a \in \mathcal{A}$, $\Gamma(s_h, a) \in \Delta(\mathcal{S}_{h+1})$. Thus, the environment transitions from state space $\mathcal{S}_1$ up to $\mathcal{S}_H$ via a sequence of actions. Layered structure allows us to avoid indexing policies and $Q$-functions with time, which enables concise notation.

Each episode produces a full record of interaction $(s_1, x_1, a_1, r_1, \ldots, s_H, x_H, a_H, r_H)$ where $s_1 \sim \Gamma_1$, $s_h \sim \Gamma(s_{h-1}, a_{h-1})$, $(x_h, r_h) \sim D_{s_h}$ and all actions $a_h$ are chosen by the learning agent. The record of interaction observed by the learner is $(x_1, a_1, r_1(a_1), \ldots, x_H, a_H, r_H(a_H))$ and at time point $h$, the learner may use all observable information up to and including $x_h$ to select $a_h$. Notice that all state information and rewards for alternative actions are unobserved by the learning agent.

The learner's reward for an episode is $\sum_{h=1}^{H} r_h(a_h)$, and the goal is to maximize the expected cumulative reward, $R = \mathbb{E}[\sum_{h=1}^{H} r_h(a_h)]$, where the expectation accounts for all the randomness in the model and the learner. We assume that almost surely $\sum_{h=1}^{H} r_h(a_h) \in [0,1]$ for any action sequence.

In this model, the optimal expected reward achievable can be computed recursively as

$$V^\star \triangleq \mathbb{E}_{s \sim \Gamma_1}[V^\star(s)] \quad \text{with} \quad V^\star(s) \triangleq \mathbb{E}_{x \sim D_s} \max_a \mathbb{E}_{r \sim D_{s|x}} \left[ r(a) + \mathbb{E}_{s' \sim \Gamma(s,a)} V^\star(s') \right]. \quad (1)$$

As the base case, we assume that for states $s \in \mathcal{S}_H$, all actions transition to a terminal state $s_{H+1}$ with $V^\star(s_{H+1}) \triangleq 0$. For each $(s, x)$ pair such that $D_s(x) > 0$ we also define a $Q^\star$ function as

$$Q_s^\star(x, a) \triangleq \mathbb{E}_{r \sim D_{s|x}} \left[ r(a) + \mathbb{E}_{s' \sim \Gamma(s,a)} V^\star(s') \right]. \tag{2}$$

This function captures the optimal choice of action given this (state, observation) pair and therefore encodes optimal behavior in the model.

With no further assumptions, the above model is a *layered episodic Partially Observable Markov Decision Process* (LE-POMDP). Both learning and planning are notoriously challenging in POMDPs, because the optimal policy depends on the entire trajectory and the complexity of learning such a policy grows exponentially with $H$ (see e.g. Kearns et al. [15] as well as Propositions 1 and 2 below). Our model avoids this statistical barrier with two assumptions: (a) we consider only reactive policies, and (b) we assume access to a class of functions that can realize the $Q^\star$ function. Both assumptions are implicit in the empirical state of the art RL results. They also eliminate issues related to partial observability, allowing us to focus on our core goal of systematic exploration. We describe both assumptions in detail before formally defining the model.

**Reactive Policies:** One approach taken by some prior theoretical work is to consider *reactive* (or memoryless) policies that use only the current observation to select an action [4, 20]. Memorylessness is slightly generalized in the recent empirical advances in RL, which typically employ policies that depend only on the few most recent observations [21].

A reactive policy $\pi : \mathcal{X} \to \mathcal{A}$ is a strategy for navigating the search space by taking actions $\pi(x)$ given observation $x$. The expected reward for a policy is defined recursively through

$$V(\pi) \triangleq \mathbb{E}_{s \sim \Gamma_1}[V(s, \pi)] \quad \text{and} \quad V(s, \pi) \triangleq \mathbb{E}_{(x,r) \sim D_s} \left[ r(\pi(x)) + \mathbb{E}_{s' \sim \Gamma(s,\pi(x))} V(s', \pi) \right].$$

A natural learning goal is to identify a policy with maximal value $V(\pi)$ from a given collection of reactive policies $\Pi$. Unfortunately, even when restricting to reactive policies, learning in POMDPs requires exponentially many samples, as we show in the next lower bound.

**Proposition 1.** *Fix $H, K \in \mathbb{N}$ with $K \geq 2$ and $\epsilon \in (0, \sqrt{1/8})$. For any algorithm, there exists a LE-POMDP with horizon $H$, $K$ actions, and $2H$ total states; a class $\Pi$ of reactive policies with $|\Pi| = K^H$; and a constant $c > 0$ such that the probability that the algorithm outputs a policy $\hat{\pi}$ with $V(\hat{\pi}) > \max_{\pi \in \Pi} V(\pi) - \epsilon$ after collecting $T$ trajectories is at most $2/3$ for all $T \leq cK^H / \epsilon^2$.*

This lower bound precludes a $\text{Poly}(K, M, H, \log(|\Pi|))$ sample complexity bound for learning reactive policies in general POMDPs as $\log(|\Pi|) = H \log(K)$ in the construction, but the number of samples required is exponential in $H$. The lower bound instance provides essentially no instantaneous feedback and therefore forces the agent to reason over $K^H$ paths independently.

**Predictability of $Q^\star$:** The assumption underlying the empirical successes in RL is that the $Q^\star$ function can be well-approximated by some large set of functions $\mathcal{F}$. To formalize this assumption, note that for some POMDPs, we may be able to write $Q^\star$ as a function of the observed history $(x_1, a_1, r_1(a_1), \ldots, x_h)$ at time $h$. For example, this is always true in deterministic-transition POMDPs, since the sequence of previous actions encodes the state and $Q^\star$ as in Eq. (2) depends only on the state, the current observation, and the proposed action. In the *realizable* setting, we have access to a collection of functions $\mathcal{F}$ mapping the observed history to $[0, 1]$, and we assume that $Q^\star \in \mathcal{F}$.

Unfortunately, even with realizability, learning in POMDPs can require exponentially many samples.

**Proposition 2.** *Fix $H, K \in \mathbb{N}$ with $K \geq 2$ and $\epsilon \in (0, \sqrt{1/8})$. For any algorithm, there exists a LE-POMDP with time horizon $H$, $K$ actions, and $2H$ total states; a class of predictors $\mathcal{F}$ with $|\mathcal{F}| = K^H$ and $Q^\star \in \mathcal{F}$; and a constant $c \geq 0$ such that the probability that the algorithm outputs a policy $\hat{\pi}$ with $V(\hat{\pi}) > V^\star - \epsilon$ after collecting $T$ trajectories is at most $2/3$ for all $T \leq cK^H / \epsilon^2$.*

As with Proposition 1, this lower bound precludes a $\text{Poly}(K, M, H, \log(|\Pi|))$ sample complexity bound for learning POMDPs with realizability. The lower bound shows that even with realizability, the agent may have to reason over $K^H$ paths independently since the functions can depend on the entire history. Proofs of both lower bounds here are deferred to Appendix A.

Both lower bounds use POMDPs with deterministic transitions and an extremely small observation space. Consequently, even learning in deterministic-transition POMDPs requires further assumptions.

## 2.2 Main Assumptions

As we have seen, neither restricting to reactive policies, nor imposing realizability enable tractable learning in POMDPs on their own. Combined however, we will see that sample-efficient learning is possible, and the combination of these two assumptions is precisely how we characterize our model. Specifically, we study POMDPs for which $Q^\star$ can be realized by a predictor that uses only the current observation and proposed action.

**Assumption 1** (*Reactive Value Functions*)**.** We assume that for all $x \in \mathcal{X}, a \in \mathcal{A}$ and any two state $s, s'$ such that $D_s(x), D_{s'}(x) > 0$, we have $Q^\star_s(x, a) = Q^\star_{s'}(x, a)$.

The restriction on $Q^\star$ implies that the optimal policy is reactive and also that the optimal predictor of long-term reward depends only on the current observation. In the following section, we describe how this condition relates to other RL models in the literature. We first present a natural example.

**Example 1** (*Disjoint observations*)**.** The simplest example is one where each state $s$ can be identified with a subset $\mathcal{X}_s$ with $D_s(x) > 0$ only for $x \in \mathcal{X}_s$ and where $\mathcal{X}_s \cap \mathcal{X}_{s'} = \emptyset$ when $s \neq s'$. A realized observation then uniquely identifies the underlying state $s$ so that Assumption 1 trivially holds, but this mapping from $s$ to $\mathcal{X}_s$ is unknown to the agent. Thus, the problem cannot be easily reduced to a small-state MDP. This setting is quite natural in several robotics and navigation tasks, where the visual signals are rich enough to uniquely identify the agent's position (and hence state). It also applies to video game playing, where the raw pixel intensities suffice to decode the game's memory state, but learning this mapping is challenging.

Thinking of $x$ as the state, the above example is an MDP with infinite state space but with structured transition operator. While our model is more general, we are primarily motivated by these infinite-state MDPs, for which the reactivity assumptions are completely non-restrictive. For infinite-state MDPs, our model describes a particular structure on the transition operator that we show enables efficient learning. We emphasize that our focus is not on partial observability issues.

As we are interested in understanding function approximation, we make a realizability assumption.

**Assumption 2** (*Realizability*)**.** We are given access to a class of predictors $\mathcal{F} \subseteq (\mathcal{X} \times \mathcal{A} \to [0, 1])$ of size $|\mathcal{F}| = N$ and assume that $Q^\star = f^\star \in \mathcal{F}$. We identify each predictor $f$ with a policy $\pi_f(x) \triangleq \operatorname{argmax}_a f(x, a)$. Observe that the optimal policy is $\pi_{f^\star}$ which satisfies $V(\pi_{f^\star}) = V^\star$.

Assumptions 1 and 2 exclude the lower bounds from Propositions 1 and 2. Our algorithm requires one further assumption.

**Assumption 3** (*Deterministic Transitions*)**.** We assume that the transition model is deterministic. This means that the starting distribution $\Gamma_1$ is a point-mass on some state $s_1$ and $\Gamma : (\mathcal{S} \times \mathcal{A}) \to \mathcal{S}$.

Even with deterministic transitions, learning requires systematic global exploration that is unaddressed in previous work. Recall that the lower bound constructions for Propositions 1 and 2 actually use deterministic transition POMDPs. Therefore, deterministic transitions combined with either the reactive or the realizability assumption by itself still precludes tractable learning. Nevertheless, we hope to relax this final assumption in future work.

More broadly, this model provides a framework to reason about reinforcement learning with function approximation. This is highly desirable as such approaches are the empirical state-of-the-art, but the limited supporting theory provides little advice on systematic global exploration.

## 2.3 Connections to Other Models and Techniques

The above model is closely related to several well-studied models in the literature, namely:

**Contextual Bandits:** If $H = 1$, then our model reduces to stochastic contextual bandits [8, 16], a well-studied simplification of the general reinforcement learning problem. The main difference is that the choice of action *does not* influence the future observations (there is only one state), and algorithms do not need to perform long-term planning to obtain low sample complexity.

**Markov Decision Processes:** If $\mathcal{X} = \mathcal{S}$ and $D_s(x)$ for each state $s$ is concentrated on $s$, then our model reduces to small-state MDPs, which can be efficiently solved by tabular approaches [6, 13, 26]. The key differences in our setting are that the observation space $\mathcal{X}$ is extremely large or infinite

and the underlying state is unobserved, so tabular methods are not viable and algorithms need to *generalize* across observations.

When the number of states is large, existing methods typically require exponentially many samples such as the $\mathcal{O}(K^H)$ result of Kearns et al. [15]. Others depend poorly on the complexity of the policy set or scale linearly in the size of a covering over the state space [10, 12, 23]. Lastly, policy gradient methods avoid dependence on size of the state space, but do not achieve global optimality [11, 27] in theory and in practice, unlike our algorithm which is guaranteed to find the globally optimal policy.

**POMDPs:** By definition our model is a POMDP where the $Q^\star$ function is consistent across states. This restriction implies that the agent does not have to reason over belief states as is required in POMDPs. There are some sample complexity guarantees for learning in arbitrarily complex POMDPs, but the bounds we are aware of are quite weak as they scale linearly with $|\Pi|$ [14, 19], or require discrete observations from a small set [4].

**State Abstraction:** State abstraction (see [18] for a survey) focuses on understanding what optimality properties are preserved in an MDP after the state space is compressed. While our model does have a small number of underlying states, they do not necessarily admit non-trivial state abstractions that are easy to discover (i.e. that do not amount to learning the optimal behavior) as the optimal behavior can depend on the observation in an arbitrary manner. Furthermore, most sample complexity results cannot search over large abstraction sets (see e.g. Jiang et al. [9]), limiting their scope.

**Function Approximation:** Our approach uses function approximation to address the generalization problem implicit in our model. Function approximation is the empirical state-of-the-art in reinforcement learning [21], but theoretical analysis has been quite limited. Several authors have studied linear or more general function approximation (See [5, 24, 28]), but none of these results give finite sample bounds, as they do not address the exploration question. Li and Littman [17] do give finite sample bounds, but they assume access to a "Knows-what-it-knows" (KWIK) oracle, which cannot exist even for simple problems. Other theoretical results either make stronger realizability assumptions (c.f., [2]) or scale poorly with problem parameters (e.g., polynomial in the number of functions [22] or the size of the observation space [23]).

# 3 The Result

We consider the task of Probably Approximately Correct (PAC) learning the models defined in Section 2. Given $\mathcal{F}$ (Assumption 2), we say that an algorithm PAC learns our model if for any $\epsilon, \delta \in (0, 1)$, the algorithm outputs a policy $\hat{\pi}$ satisfying $V(\hat{\pi}) \geq V^\star - \epsilon$ with probability at least $1 - \delta$. The *sample complexity* is a function $n : (0, 1)^2 \to \mathbb{N}$ such that for any $\epsilon, \delta \in (0, 1)$, the algorithm returns an $\epsilon$-suboptimal policy with probability at least $1 - \delta$ using at most $n(\epsilon, \delta)$ episodes. We refer to a $\text{Poly}(M, K, H, 1/\epsilon, \log N, \log(1/\delta))$ sample complexity bound as polynomial in all relevant parameters. Notably, there should be no dependence on $|\mathcal{X}|$, which may be infinite.

## 3.1 The Algorithm

Before turning to the algorithm, it is worth clarifying some additional notation. Since we are focused on the deterministic transition setting, it is natural to think about the environment as an exponentially large search tree with fan-out $K$ and depth $H$. Each node in the search tree is labeled with an (unobserved) state $s \in \mathcal{S}$, and each edge is labeled with an action $a \in \mathcal{A}$, consistent with the transition model. A path $p \in \mathcal{A}^\star$ is a sequence of actions from the root of the search tree, and we also use $p$ to denote the state reached after executing the path $p$ from the root. Thus, $D_p$ is the observation distribution of the state at the end of the path $p$. We use $p \circ a$ to denote a path formed by executing all actions in $p$ and then executing action $a$, and we use $|p|$ to denote the length of the path. Let $\varnothing$ denote the empty path, which corresponds to the root of the search tree.

The pseudocode for the algorithm, which we call Least Squares Value Elimination by Exploration (LSVEE), is displayed in Algorithm 1 (See also Appendix B). LSVEE has two main components: a depth-first-search routine with a learning step (step 6 in Algorithm 2) and an on-demand exploration technique (steps 5-8 in Algorithm 1). The high-level idea of the algorithm is to eliminate regression functions that do not meet Bellman-like consistency properties of the $Q^\star$ function. We now describe both components and their properties in detail.

---

**Algorithm 1** Least Squares Value Elimination by Exploration: LSVEE $(\mathcal{F}, \epsilon, \delta)$

---
1: $\mathcal{F} \leftarrow$ DFS-LEARN$(\varnothing, \mathcal{F}, \epsilon, \delta/2)$.
2: Choose any $f \in \mathcal{F}$. Let $\hat{V}^{\star}$ be a Monte Carlo estimate of $V^f(\varnothing, \pi_f)$. (See Eq. (3))
3: Set $\epsilon_{\text{demand}} = \epsilon/2, n_1 = \frac{32 \log(12MH/\delta)}{\epsilon^2}$ and $n_2 = \frac{8 \log(6MH/\delta)}{\epsilon}$.
4: **while** true **do**
5:     Fix a regressor $f \in \mathcal{F}$.
6:     Collect $n_1$ trajectories according to $\pi_f$ and estimate $V(\pi_f)$ via Monte-Carlo estimate $\hat{V}(\pi_f)$.
7:     If $|\hat{V}(\pi_f) - \hat{V}^{\star}| \leq \epsilon_{\text{demand}}$, return $\pi_f$.
8:     Otherwise update $\mathcal{F}$ by calling DFS-LEARN $(p, \mathcal{F}, \epsilon, \frac{\delta}{6MH^2 n_2})$ on each of the $H - 1$ prefixes $p$ of each of the first $n_2$ paths collected in step 6.
9: **end while**

---

---

**Algorithm 2** DFS-LEARN $(p, \mathcal{F}, \epsilon, \delta)$

---
1: Set $\phi = \frac{\epsilon}{320 H^2 \sqrt{K}}$ and $\epsilon_{\text{test}} = 20(H - |p| - 5/4)\sqrt{K}\phi$.
2: **for** $a \in \mathcal{A}$, if not CONSENSUS$(p \circ a, \mathcal{F}, \epsilon_{\text{test}}, \phi, \frac{\delta/2}{MKH})$ **do**
3:     $\mathcal{F} \leftarrow$ DFS-LEARN$(p \circ a, \mathcal{F}, \epsilon, \delta)$.
4: **end for**
5: Collect $n_{\text{train}} = \frac{24}{\phi^2} \log\left(\frac{8MHN}{\delta}\right)$ observations $(x_i, a_i, r_i)$ where $(x_i, r_i') \sim D_p$, $a_i$ is chosen uniformly at random, and $r_i = r_i'(a_i)$.
6: Return $\left\{ f \in \mathcal{F} : \tilde{R}(f) \leq \min_{f' \in \mathcal{F}} \tilde{R}(f') + 2\phi^2 + \frac{22 \log(4MHN/\delta)}{n_{\text{train}}} \right\}$, $\tilde{R}(f)$ defined in Eq. (4).

---

**The DFS routine**: When the DFS routine, displayed in Algorithm 2, is run at some path $p$, we first decide whether to recursively expand the descendants $p \circ a$ by performing a *consensus test*. Given a path $p'$, this test, displayed in Algorithm 3, computes estimates of *value predictions*,

$$V^f(p', \pi_f) \triangleq \mathbb{E}_{x \sim D_{p'}} f(x, \pi_f(x)), \tag{3}$$

for all the surviving regressors. These value predictions are easily estimated by collecting many observations after rolling in to $p'$ and using empirical averages (See line 2 in Algorithm 3). If all the functions agree on this value for $p'$ the DFS need not visit this path.

After the recursive calls, the DFS routine performs the *elimination step* (line 6). When this step is invoked at path $p$, the algorithm collects $n_{\text{train}}$ observations $(x_i, a_i, r_i)$ where $(x_i, r_i') \sim D_p$, $a_i$ is chosen uniformly at random, and $r_i = r_i'(a_i)$ and eliminates regressors that have high empirical risk,

$$\tilde{R}(f) \triangleq \frac{1}{n_{\text{train}}} \sum_{i=1}^{n_{\text{train}}} (f(x_i, a_i) - r_i - \hat{V}^f(p \circ a_i, \pi_f))^2. \tag{4}$$

**Intuition for DFS**: This regression problem is motivated by the realizability assumption and the definition of $Q^\star$ in Eq. (2), which imply that at path $p$ and for all actions $a$,

$$f^\star(x, a) = \mathbb{E}_{r \sim D_{p|x}} r(a) + V(p \circ a, \pi_{f^\star}) = \mathbb{E}_{r \sim D_{p|x}} r(a) + \mathbb{E}_{x' \sim D_{p \circ a}} f^\star(x', \pi_{f^\star}(x')). \tag{5}$$

Thus $f^\star$ is consistent between its estimate at the current state $s$ and the future state $s' = \Gamma(s, a)$.

The regression problem (4) is essentially a finite sample version of this identity. However, some care must be taken as the target for the regression function $f$ includes $V^f(p \circ a, \pi_f)$, which is $f$'s value prediction for the future. The fact that the target differs across functions can cause instability in the regression problem, as some targets may have substantially lower variance than $f^\star$'s. To ensure correct behavior, we must obtain high-quality future value prediction estimates, and so, we re-use the Monte-Carlo estimates $\hat{V}^f(p \circ a, \pi_f)$ in Eq. (3) from the consensus tests. Each time we perform elimination, the regression targets are close for all considered $f$ in Equation (4) owing to consensus being satisfied at the successor nodes in Step 2 of Algorithm 2.

Given consensus at all the descendants, each elimination step inductively propagates learning towards the start state by ensuring the following desirable properties hold: (i) $f^\star$ is not eliminated, (ii)

---

**Algorithm 3** CONSENSUS$(p, \mathcal{F}, \epsilon_{\text{test}}, \phi, \delta)$

---

1: Set $n_{\text{test}} = \frac{2}{\phi^2} \log(2N/\delta)$. Collect $n_{\text{test}}$ observations $x_i \sim D_p$.

2: Compute for each function, $\hat{V}^f(p, \pi_f) = \frac{1}{n_{\text{test}}} \sum_{i=1}^{n_{\text{test}}} f(x_i, \pi_f(x_i))$.

3: Return $\mathbf{1} \left[ |\hat{V}^f(p, \pi_f) - \hat{V}^g(p, \pi_g)| \le \epsilon_{\text{test}} \; \forall f, g \in \mathcal{F} \right].$

---

consensus is reached at $p$, and (iii) surviving policies choose good actions at $p$. Property (ii) controls the sample complexity, since consensus tests at state $s$ return true once elimination has been invoked on $s$, so DFS avoids exploring the entire search space. Property (iii) leads to the PAC-bound; if we have run the elimination step on all states visited by a policy, that policy must be near-optimal.

To bound the sample complexity of the DFS routine, since there are $M$ states per level and the consensus test returns true once elimination has been performed, we know that the DFS does not visit a large fraction of the search tree. Specifically, this means DFS is invoked on at most $MH$ nodes in total, so we run elimination at most $MH$ times, and we perform at most $MKH$ consensus tests. Each of these operations requires polynomially many samples.

The elimination step is inspired by the RegressorElimination algorithm of Agarwal et. al [1] for contextual bandit learning in the realizable setting. In addition to forming a different regression problem, RegressorElimination carefully chooses actions to balance exploration and exploitation which leads to an optimal regret bound. In contrast, we are pursuing a PAC-guarantee here, for which it suffices to focus exclusively on exploration.

**On-demand Exploration**: While DFS is guaranteed to estimate the optimal value $V^\star$, it unfortunately does not identify the optimal policy. For example, if consensus is satisfied at a state $s$ without invoking the elimination step, then each function accurately predicts the value $V^\star(s)$, but the associated policies are not guaranteed to achieve this value. To overcome this issue, we use an *on-demand exploration* technique in the second phase of the algorithm (Algorithm 1, steps 5-8).

At each iteration of this phase, we select a policy $\pi_f$ and estimate its value via Monte Carlo sampling. If the policy has sub-optimal value, we invoke the DFS procedure on many of the paths visited. If the policy has near-optimal value, we have found a good policy, so we are done. This procedure requires an accurate estimate of the optimal value, which we already obtained by invoking the DFS routine at the root, since it guarantees that all surviving regressors agree with $f^\star$'s value on the starting state distribution. $f^\star$'s value is precisely the optimal value.

**Intuition for On-demand Exploration**: Running the elimination step at some path $p$ ensures that all surviving regressors take good actions at $p$, in the sense that taking one action according to any surviving policy and then behaving optimally thereafter achieves near-optimal reward for path $p$. This does not ensure that all surviving policies achieve near-optimal reward, because they may take highly sub-optimal actions after the first one. On the other hand, if a surviving policy $\pi_f$ visits only states for which the elimination step has been invoked, then it must have near-optimal reward. More precisely, letting $L$ denote the set of states for which the elimination step has been invoked (the "learned" states), we prove that any surviving $\pi_f$ satisfies

$$V^\star - V(\pi_f) \le \epsilon/8 + \mathbb{P}\left[\pi_f \text{ visits a state } s \notin L\right]$$

Thus, if $\pi_f$ is highly sub-optimal, it must visit some unlearned states with substantial probability. By calling DFS-LEARN on the paths visited by $\pi_f$, we ensure that the elimination step is run on at least one unlearned states. Since there are only $MH$ distinct states and each non-terminal iteration ensures training on an unlearned state, the algorithm must terminate and output a near-optimal policy.

Computationally, the running time of the algorithm may be $O(N)$, since eliminating regression functions according to Eq. (4) may require enumerating over the class and the consensus function requires computing the maximum and minimum of $N$ numbers, one for each function. This may be intractably slow for rich function classes, but our focus is on statistical efficiency, so we ignore computational issues here.

### 3.2 The PAC Guarantee

Our main result certifies that LSVEE PAC-learns our models with polynomial sample complexity.

**Theorem 1** (PAC bound). *For any $(\epsilon, \delta) \in (0, 1)$ and under Assumptions 1, 2, and 3, with probability at least $1 - \delta$, the policy $\pi$ returned by* LSVEE *is at most $\epsilon$-suboptimal. Moreover, the number of episodes required is at most*

$$\tilde{\mathcal{O}} \left( \frac{MH^6K^2}{\epsilon^3} \log(N/\delta) \log(1/\delta) \right).$$

This result uses the $\tilde{\mathcal{O}}$ notation to suppress logarithmic dependence in all parameters except for $N$ and $\delta$. The precise dependence on all parameters can be recovered by examination of our proof and is shortened here simply for clarity. See Appendix C for the full proof of the result.

This theorem states that LSVEE produces a policy that is at most $\epsilon$-suboptimal using a number of episodes that is polynomial in all relevant parameters. To our knowledge, this is the first polynomial sample complexity bound for reinforcement learning with infinite observation spaces, without prohibitively strong assumptions (e.g., [2, 22, 23]). We also believe this is the first finite-sample guarantee for reinforcement learning with general function approximation without prohibitively strong assumptions (e.g., [2]).

Since our model generalizes both contextual bandits and MDPs, it is worth comparing the sample complexity bounds.

1.  In contextual bandits, we have $M = H = 1$ so that the sample complexity of LSVEE is $\tilde{\mathcal{O}}(\frac{K^2}{\epsilon^3} \log(N/\delta) \log(1/\delta))$, in contrast with known $\tilde{\mathcal{O}}(\frac{K}{\epsilon^2} \log(N/\delta))$ results.

2.  Prior results establish the sample complexity for learning layered episodic MDPs with deterministic transitions is $\tilde{\mathcal{O}}(\frac{MK\text{poly}(H)}{\epsilon^2} \log(1/\delta))$ [7, 25].

Both comparisons show our sample complexity bound may be suboptimal in its dependence on $K$ and $\epsilon$. Looking into our proof, the additional factor of $K$ comes from collecting observations to estimate the value of future states, while the additional $1/\epsilon$ factor arises from trying to identify a previously unexplored state. In contextual bandits, these issues do not arise since there is only one state, while, in tabular MDPs, they can be trivially resolved as the states are observed. Thus, with minor modifications, LSVEE can avoid these dependencies for both special cases. In addition, our bound disagrees with the MDP results in the dependence on the policy complexity $\log(N)$; which we believe is unavoidable when working with rich observation spaces.

Finally, our bound depends on the number of states $M$ in the worst case, but the algorithm actually uses a more refined notion. Since the states are unobserved, the algorithm considers two states distinct only if they have reasonably different value functions, meaning learning on one does not lead to consensus on the other. Thus, a more distribution-dependent analysis defining states through the function class is a promising avenue for future work.

## 4 Discussion

This paper introduces a new model in which it is possible to design and analyze principled reinforcement learning algorithms engaging in global exploration. As a first step, we develop a new algorithm and show that it learns near-optimal behavior under a deterministic-transition assumption with polynomial sample complexity. This represents a significant advance in our understanding of reinforcement learning with rich observations. However, there are major open questions:

1.  Do polynomial sample bounds for this model with stochastic transitions exist?

2.  Can we design an algorithm for learning this model that is both computationally and statistically efficient? The sample complexity of our algorithm is logarithmic in the size of the function class $\mathcal{F}$ but uses an intractably slow enumeration of these functions.

Good answers to both of these questions may yield new practical reinforcement learning algorithms.

## Acknowledgements

We thank Akshay Balsubramani and Hal Daumé III for formative discussions, and we thank Tzu-Kuo Huang and Nan Jiang for carefully reading an early draft of this paper. This work was carried out while AK was at Microsoft Research.

## Footnotes

[1]We use "global exploration" to distinguish the sophisticated exploration strategies required to solve an MDP efficiently from exponentially less efficient alternatives such as $\epsilon$-greedy.

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
