[Supplementary Material]

# A  The Lower Bounds

**Theorem 2** (Lower bound for best arm identification in stochastic bandits). *For any $K \geq 2$ and $\epsilon \leq \sqrt{1/8}$ and any best-arm identification algorithm, there exists a multi-armed bandit problem for which the best arm $i^\star$ is $\epsilon$ better than all others, but for which the estimate $\hat{i}$ of the best arm must have $\mathbb{P}[\hat{i} \neq i^\star] \geq 1/3$ unless the number of samples collected $T$ is at least $\frac{K}{72\epsilon^2}$.*

*Proof.* The proof is essentially the same as the regret lower bound for stochastic multi-armed bandits from Auer et al. [3]. Since we want the lower bound for best arm identification instead of regret, we include a full proof for completeness.

Following Auer et al. [3], the lower bound instance is drawn uniformly from a family of multi-armed bandit problems with $K$ arms each. There are $K$ problems in the family, and each one is parametrized by the optimal arm $i^\star$. For the $i^{\star\text{th}}$ problem, arm $i^\star$ produces rewards drawn from $\text{Ber}(1/2 + \epsilon)$ while all other arms produce rewards from $\text{Ber}(1/2)$. Let $\mathbb{P}_{i^\star}$ denote the reward distribution for the $i^{\star\text{th}}$ bandit problem, so that $\mathbb{P}_{i^\star}(\cdot | a = i^\star) = \text{Ber}(1/2 + \epsilon)$ and $\mathbb{P}_{i^\star}(\cdot | a \neq i^\star) = \text{Ber}(1/2)$. Let $\mathbb{P}_0$ denote the reward distribution where all arms receive $\text{Ber}(1/2)$ rewards.

Since the environment is stochastic, any randomized algorithm is just a distribution over deterministic ones, and it therefore suffices to consider only deterministic algorithms. More precisely, a randomized algorithm uses some random bits $z$ and for each choice, the algorithm itself is deterministic. If we lower bound $\mathbb{P}_{i^\star}[\hat{i} \neq i^\star | z]$ for all $z$, then we also obtain a lower bound after taking expectation.

A deterministic algorithm can be specified as a sequence of mappings $\psi_t : \{0, 1\}^t \to [K]$ with the interpretation of $\psi_T$ as the estimate of the best arm. Note that $\psi_0$ is the first arm chosen, which does not depend on any of the observations. The algorithm can be specified this way since the sequence of actions played can be inferred by the sequence of observed rewards. Let $\mathbb{P}_{i^\star, \psi}$ denote the distribution over all $T$ rewards when $i^\star$ is the optimal arm and actions are selected according to $\psi$. We are interested in bounding the error event $\mathbb{P}_{i^\star, \psi}[\psi_T \neq i^\star]$.

We first prove,

$$\mathbb{P}_{i^\star, \psi}[\psi_T = i^\star] - \mathbb{P}_{0, \psi}[\psi_T = i^\star] \leq \frac{1}{2}\sqrt{\mathbb{E}_{0, \psi}[N_{i^\star}] \log \frac{1}{1 - 4\epsilon^2}},$$

where $N_i$ is the number of times $\psi$ plays action $i$ over the course of $T$ rounds. $N_i$ is a random variable since it depends on the sequence of observations, and here we take expectation with respect to $\mathbb{P}_0$.

To prove this statement, notice that,

$$|\mathbb{P}_{i^\star, \psi}[\psi_T = i^\star] - \mathbb{P}_{0, \psi}[\psi_T = i^\star]| \leq \|P_{i^\star, \psi} - P_{0, \psi}\|_{\text{TV}} \leq \sqrt{\frac{1}{2} KL(P_{0, \psi} || P_{i^\star, \psi})}.$$

The first inequality is by definition of the total variation distance, while the second is by Pinsker's inequality. We are left to bound the KL divergence. To do so, we introduce notation for sequences. For any $t \in \mathbb{N}$, we use $r_{1:t} \in \{0, 1\}^t$ to denote the binary reward sequence of length $t$. The KL divergence is

$$
\begin{aligned}
KL(P_{0, \psi} || P_{i^\star, \psi}) &= \sum_{r_{1:T} \in \{0,1\}^T} P_{0, \psi}(r_{1:T}) \log\left(\frac{P_{0, \psi}(r_{1:T})}{P_{i^\star, \psi}(r_{1:T})}\right) \\
&= \sum_{t=1}^{T} \sum_{r_{1:t} \in \{0,1\}^t} P_{0, \psi}(r_{1:t}) \log\left(\frac{P_{0, \psi}(r_t | r_{1:t-1})}{P_{i^\star, \psi}(r_t | r_{1:t-1})}\right) \\
&= \sum_{t=1}^{T} \sum_{r_{1:t-1} : a_t = i^\star} P_{0, \psi}(r_{1:t-1}) \left(\sum_{x \in \{0,1\}} P_{0, \psi}(x) \log\left(\frac{P_{0, \psi}(x | a_t = i^\star)}{P_{i^\star, \psi}(x | a_t = i^\star)}\right)\right),
\end{aligned}
$$

where $a_t$ is the chosen action at time $t$. To arrive at the second line we use the chain rule for KL-divergence. The third line is based on the fact that if $a_t \neq i^\star$, then the log ratio is zero, since the

two conditional distributions are identical. Continuing with straightforward calculations, we have

$$KL(P_{0,\psi}||P_{i^\star,\psi}) = \sum_{t=1}^{T} \sum_{r_{1:t-1}:a_t=i^\star} P_{0,\psi}(r_{1:t-1}) \left( \frac{1}{2} \log \left( \frac{1/2}{1/2-\epsilon} \right) + \frac{1}{2} \log \left( \frac{1/2}{1/2+\epsilon} \right) \right)$$

$$= \left( -\frac{1}{2} \log(1-4\epsilon^2) \right) \sum_{t=1}^{T} \sum_{r_{1:t-1}:a_t=i^\star} P_{0,\psi}(r_{1:t-1})$$

$$= \left( -\frac{1}{2} \log(1-4\epsilon^2) \right) \sum_{t=1}^{T} \mathbb{P}_{0,\psi}[a_t = i^\star].$$

This proves the sub-claim, which follows the same argument as as Auer et. al [3].

To prove the final result, we take expectation over the problem $i^\star$.

$$\frac{1}{K} \sum_{i^\star=1}^{K} \mathbb{P}_{i^\star,\psi}[\psi_T = i^\star] \leq \frac{1}{K} \sum_{i^\star=1}^{K} \mathbb{P}_{0,\psi}[\psi_T = i^\star] + \frac{1}{2K} \sum_{i^\star=1}^{K} \sqrt{\mathbb{E}_{0,\psi}[N_{i^\star}] \log \frac{1}{1-4\epsilon^2}}$$

$$\leq \frac{1}{K} + \frac{1}{2} \sqrt{\frac{-\log(1-4\epsilon^2)}{K} \mathbb{E}_{0,\psi} \sum_{i^\star=1}^{K} N_{i^\star}} \leq \frac{1}{K} + \frac{1}{2} \sqrt{\frac{-\log(1-4\epsilon^2)T}{K}}.$$

If $4\epsilon^2 \leq 1/2$ then $-\log(1-4\epsilon^2) \leq 8\epsilon^2$. This follows by the Taylor expansion of $-\log(1-x)$,

$$-\log(1-x) = \sum_{i=1}^{\infty} \frac{x^i}{i} \leq x \left( \sum_{i=0}^{\infty} \frac{2^{-i}}{i+1} \right) \leq x \sum_{i=0}^{\infty} 2^{-i} = 2x.$$

The inequality here uses the assumption that $x \leq 1/2$.

Thus, whenever $\epsilon \leq \sqrt{1/8}$ and $T \leq \frac{K}{72\epsilon^2}$, this number is smaller than $2/3$, since we restrict to the cases where $K \geq 2$. This is the success probability, so the failure probability is at least $1/3$, which proves the result. □

## A.1 The construction

Here we design a family of POMDPs for both lower bounds. As with multi-armed bandits above, the lower bound will be realized by sampling a POMDP from a uniform distribution over this family of problems. Fix $H$ and $K$ and pick a single $x_h \in \mathcal{X}$ for each level $h \in [H]$ so that $x_h \neq x_{h'}$ for all pairs $h \neq h'$. For each level there are two states $g_h$ and $b_h$ for "good" and "bad." The observation marginal distribution $D_{g_h} = D_{b_h}$ is concentrated on $x_h$ for each level $h$, so the observations provide no information about the underlying state. Rewards for all levels except for $h = H$ are zero.

Each POMDPs in the family corresponds to a path $p^\star = (a_1^\star, \ldots, a_H^\star) \in K^H$. The transition function for the POMDP corresponding to the path $p^\star$ is,

$$\Gamma(g_h, a_h^\star) \triangleq g_{h+1}$$
$$\Gamma(g_h, a) \triangleq b_{h+1} \text{ if } a \neq a_h^\star$$
$$\Gamma(b_h, a) \triangleq b_{h+1} \forall a.$$

The reward is drawn from $\text{Ber}(1/2 + \epsilon)$ if the last state is $g_H$ and if the last action is $a_H^\star$. For all other outcomes the reward is drawn from $\text{Ber}(1/2)$. Observe that these models have deterministic transitions.

Clearly all of the models in this family are distinct, and there are $K^H$ such models. Moreover, since the observations $x_h$ provide no information and only the final reward is non-zero, no information is received until the full sequence of actions is selected. More formally, for any two policies $\pi, \pi'$, the KL divergence between the distributions of observations and rewards produced by the two policies is exactly the KL divergence between the final rewards produced by the two policies. Therefore, the problem is equivalent to a multi-armed bandit problem with $K^H$ arms, where the optimal arm gets a $\text{Ber}(1/2 + \epsilon)$ reward while all other arms get a $\text{Ber}(1/2)$ reward. Thus, identifying a policy that is no-more than $\epsilon$ suboptimal in this POMDP is information-theoretically equivalent to identifying the best arm in the stochastic bandit problem in Theorem 2 with $K^H$ arms. Applying that lower bound gives a sample complexity bound of $\Omega(K^H/\epsilon^2)$.

## A.2 Proving both lower bounds

To verify both lower bounds in Propositions 1 and 2, we construct the policy and regressor sets. For Proposition 1, we need a set of reactive policies such that finding the optimal policy has a large sample complexity. To this end, we use the set of all $K^H$ mappings from the $H$ observations to actions. Specifically, each policy $\pi$ is identified with a sequence of $H$ actions $(a_1, \ldots, a_H)$ and has $\pi(x_h) = a_h$. These policies are reactive by definition since they do not depend on any previous history, or state of the world. Clearly there are $K^H$ such policies, and each policy is optimal for exactly one POMDP defined above, namely $\pi_p$ is optimal for the POMDP corresponding to the path $p$. Furthermore, in the POMDP defined by $p$, we have $V(\pi_p) = 1/2 + \epsilon$, whereas $V(\pi) = 1/2$ for every other policy. Consequently, finding the best policy in the class is equivalent to identifying the best arm in this family of problems. Taking a uniform mixture of problems in the family as before, we reason that this requires at least $\Omega(K^H/\epsilon^2)$ trajectories.

For Proposition 2, we use a similar construction. For each path $p = (a_1, \ldots, a_H)$, we associate a regressor $f_p$ with,

$$f_p(\rho) \triangleq \frac{1}{2} + \epsilon \mathbf{1}[\rho \text{ is a prefix of } p].$$

Here we use $\rho$ to denote the history of the interaction, which can be condensed to a sequence of actions since the observations provide no information.

Clearly for the POMDP parameterized by $p$, $f_p$ correctly maps the history to future reward, meaning that the POMDP is realizable for this regressor class. Relatedly, $\pi_{f_p}$ is the optimal policy for the POMDP with optimal sequence $p$. Moreover, there are precisely $K^H$ regressors. As before, the learning objective requires identifying the optimal policy and hence the optimal path, which requires $\Omega(K^H/\epsilon^2)$ trajectories.

# B  Full Algorithm Pseudocode

It is more natural to break the algorithm into more components for the analysis. This lets us focus on each component in isolation.

We first clarify some notation involving value functions. For predictor $f$ and policy $\pi$, we use,

$$V^f(s, \pi) \triangleq \mathbb{E}_{x \sim D_s}[f(x, \pi(x))]$$
$$V(s, \pi) \triangleq \mathbb{E}_{x \sim D_s}[r(\pi(x)) + \mathbb{E}_{s' \sim \Gamma(s, \pi(x))} V(s', \pi)].$$

Recall that $V(s_{H+1}, \pi) = 0$ for all $s_{H+1}$, which is a terminating state.

We often use a path $p$ as the first argument, with the convention that the associated state is the last one on the path. This is enabled by deterministic transitions. If a state is omitted from these functions, then it is assumed to be the start state or the root of the search tree. We also use $V^\star$ for the optimal value, where by assumption we have $V^\star = V(\pi_{f^\star}) = V^{f^\star}(\pi_{f^\star})$. Finally, throughout the algorithm and analysis, we use Monte Carlo estimates of these quantities, which we denote as $\hat{V}^f, \hat{V}$, etc.

Pseudocode for the compartmentalized version of the algorithm is displayed in Algorithm 4 with subroutines displayed as Algorithms 5, 6, 7, and 8. The algorithm should be invoked as LSVEE($\mathcal{F}, \epsilon, \delta$) where $\mathcal{F}$ is the given class of regression functions, $\epsilon$ is the target accuracy and $\delta$ is the target failure probability. The two main components of the algorithm are the DFS-LEARN and EXPLORE-ON-DEMAND routines. DFS-LEARN ensures proper invocation of the training step, TD-ELIM, by verifying a number of preconditions, while EXPLORE-ON-DEMAND finds regions of the search tree for which training must be performed.

It is easily verified that this is an identical description of the algorithm.

# C  The Full Analysis

The proof of the theorem hinges on analysis of the the subroutines. We turn first to the TD-ELIM routine, for which we show the following guarantee. Recall the definition,

$$V^f(p, \pi_f) \triangleq \mathbb{E}_{x \sim D_p} f(x, \pi_f(x)).$$

---

**Algorithm 4** Least Squares Value Elimination by Exploration: LSVEE $(\mathcal{F}, \epsilon, \delta)$

---

1: $\mathcal{F} \leftarrow$ DFS-LEARN$(\varnothing, \mathcal{F}, \epsilon, \delta/2)$.
2: Choose any $f \in \mathcal{F}$. Let $\hat{V}^\star$ be a Monte Carlo estimate of $V^f(\varnothing, \pi_f)$.
3: $f \leftarrow$ EXPLORE-ON-DEMAND$(\mathcal{F}, \hat{V}^\star, \epsilon, \delta/2)$.
4: Return $\pi_f$.

---

---

**Algorithm 5** DFS-LEARN $(p, \mathcal{F}, \epsilon, \delta)$

---

1: Set $\phi = \frac{\epsilon}{320 H^2 \sqrt{K}}$ and $\epsilon_{\text{test}} = 20(H - |p| - 5/4)\sqrt{K}\phi$.
2: **for** $a \in \mathcal{A}$ **do**
3:      **if** Not CONSENSUS$(p \circ a, \mathcal{F}, \epsilon_{\text{test}}, \phi, \frac{\delta/2}{MKH})$ **then**
4:          $\mathcal{F} \leftarrow$ DFS-LEARN$(p \circ a, \mathcal{F}, \epsilon, \delta)$.          # Recurse
5:      **end if**
6: **end for**
7: $\hat{\mathcal{F}} \leftarrow$ TD-ELIM $\left(p, \mathcal{F}, \phi, \frac{\delta/2}{MH}\right)$.          # Learn in state $p$.
8: Return $\hat{\mathcal{F}}$.

---

**Theorem 3** (Guarantee for TD-ELIM). *Consider running* TD-ELIM *at path $p$ with regressors $\mathcal{F}$, parameters $\phi, \delta$ and with $n_{train} = 24 \log(4N/\delta)/\phi^2$. Suppose that the following are true:*

1. **Estimation Precondition:** *We have access to estimates $\hat{V}^f(p \circ a, \pi_f)$ for all $f \in \mathcal{F}, a \in \mathcal{A}$ such that, $|\hat{V}^f(p \circ a, \pi_f) - V^f(p \circ a, \pi_f)| \leq \phi$.*

2. **Bias Precondition:** *For all $f, g \in \mathcal{F}$ and for all $a \in \mathcal{A}$, $|V^f(p \circ a, \pi_f) - V^g(p \circ a, \pi_g)| \leq \tau_1$.*

*Then the following hold simultaneously with probability at least $1 - \delta$:*

1. *$f^\star$ is retained by the algorithm.*
2. **Bias Bound**:
$$|V^f(p, \pi_f) - V^g(p, \pi_g)| \leq 8\phi\sqrt{K} + 2\phi + \tau_1. \tag{7}$$

3. **Instantaneous Risk Bound**:
$$V^\star(p) - V^{f^\star}(p, \pi_f) \leq 4\phi\sqrt{2K} + 2\phi + 2\tau_1. \tag{8}$$

4. **Estimation Bound**: *Regardless of whether the preconditions hold, we have estimates $\hat{V}^f(p, \pi_f)$ with,*
$$|\hat{V}^f(p, \pi_f) - V^f(p, \pi_f)| \leq \frac{\phi}{\sqrt{12}}. \tag{9}$$

*The last three bounds hold for all surviving $f, g \in \mathcal{F}$.*

The theorem shows that, as long as we call TD-ELIM with the two preconditions, then $f^\star$, the optimal regressor, always survives. It also establishes a number of other properties about the surviving functions, namely that they agree on the value of this path (the bias bound) and that the associated policies take good actions from this path (the instantaneous risk bound). Note that the instantaneous risk bound is *not* a cumulative risk bound. The second term on the left hand side is the reward achieved by behaving like $\pi_f$ for one action but then behaving optimally afterwards. The proof is deferred to Appendix E.

Analysis of the CONSENSUS subroutine requires only standard concentration-of-measure arguments.

**Theorem 4** (Guarantee for CONSENSUS). *Consider running* CONSENSUS *on path $p$ with $n_{test} = 2 \log(2N/\delta)/\phi^2$ and $\epsilon_{test} \geq 2\phi + \tau_2$, for some $\tau_2 > 0$.*

(i) *With probability at least $1 - \delta$, we have estimates $\hat{V}^f(p, \pi_f)$ with $|\hat{V}^f(p, \pi_f) - V^f(p, \pi_f)| \leq \phi \quad \forall f \in \mathcal{F}$.*

**Algorithm 6** CONSENSUS$(p, \mathcal{F}, \epsilon_{\text{test}}, \phi, \delta)$

---

Set $n_{\text{test}} = 2\log(2N/\delta)/\phi^2$.
Collect $n_{\text{test}}$ observations $x_i \sim D_p$.
Compute Monte-Carlo estimates for each value function,

$$\hat{V}^f(p, \pi_f) = \frac{1}{n_{\text{test}}} \sum_{i=1}^{n_{\text{test}}} f(x_i, \pi_f(x_i)) \qquad \forall f \in \mathcal{F}.$$

**if** $|\hat{V}^f(p, \pi_f) - \hat{V}^g(p, \pi_g)| \leq \epsilon_{\text{test}}$ for all $f, g \in \mathcal{F}$ **then**
    return `true`.
**end if**
Return `false`.

---

**Algorithm 7** TD-ELIM$(p, \mathcal{F}, \phi, \delta)$

---

Require estimates $\hat{V}^f(p \circ a, \pi_f), \forall f \in \mathcal{F}, a \in \mathcal{A}$.
Set $n_{\text{train}} = 24\log(4N/\delta)/\phi^2$.
Collect $n_{\text{train}}$ observations $(x_i, a_i, r_i)$ where $x_i \sim D_p$, $a_i$ is chosen uniformly at random, and $r_i = r_i(a_i)$.
Update $\mathcal{F}$ to

$$\left\{ f \in \mathcal{F} : \tilde{R}(f) \leq \min_{f' \in \mathcal{F}} \tilde{R}(f') + 2\phi^2 + \frac{22\log(2N/\delta)}{n_{\text{train}}} \right\},$$

$$\text{with } \tilde{R}(f) \triangleq \frac{1}{n_{\text{train}}} \sum_{i=1}^{n_{\text{train}}} (f(x_i, a_i) - r_i - \hat{V}^f(p \circ a_i, \pi_f))^2. \tag{6}$$

Return $\mathcal{F}$.

---

(ii) *If* $|V^f(p, \pi_f) - V^g(p, \pi_g)| \leq \tau_2, \forall f, g \in \mathcal{F}$, *under the event (i), the algorithm returns* `true`.

(iii) *If the algorithm returns* `true`, *then under the event in (1), we have* $|V^f(p, \pi_f) - V^g(p, \pi_g)| \leq 2\phi + \epsilon_{\text{test}} \forall f, g \in \mathcal{F}$.

Appendix F provides the proof.

Analysis of both the DFS-LEARN and EXPLORE-ON-DEMAND routines requires a careful inductive argument. We first consider the DFS-LEARN routine.

**Theorem 5** (Guarantee for DFS-LEARN). *Consider running* DFS-LEARN *on path $p$ with regressors $\mathcal{F}$, and parameters $\epsilon, \delta$. With probability at least $1 - \delta$, for all $h$ and all $s_h \in \mathcal{S}_h$ for which we called* TD-ELIM, *the conclusions of Theorem 3 hold with $\phi = \frac{\epsilon}{320 H^2 \sqrt{K}}$ and $\tau_1 = 20(H - h)\sqrt{K}\phi$. If $T$ is the number of times the algorithm calls* TD-ELIM, *then the number of episodes executed by the algorithm is at most,*

$$\mathcal{O}\left(\frac{TH^4 K^2}{\epsilon^2} \log(NMKH/\delta)\right).$$

*Moreover, $T \leq MH$ for any execution of* DFS-LEARN.

The proof details are deferred to Appendix G.

A simple consequence of Theorem 5 is that we can estimate $V^\star$ accurately once we have called DFS-LEARN on $\varnothing$.

**Corollary 1** (Estimating $V^\star$). *Consider running* DFS-LEARN *at $\varnothing$ with regressors $\mathcal{F}$, and parameters $\epsilon, \delta$. Then with probability at least $1 - \delta$, the estimate $\hat{V}^\star$ satisfies,*

$$|\hat{V}^\star - V^\star| \leq \epsilon/8.$$

*Moreover the algorithm uses at most,*

$$\mathcal{O}\left(\frac{MH^5 K^2}{\epsilon^2} \log\left(\frac{NMHK}{\delta}\right)\right)$$

---

**Algorithm 8** EXPLORE-ON-DEMAND $(\mathcal{F}, \hat{V}^\star, \epsilon, \delta)$

---

Set $\epsilon_{\text{demand}} = \epsilon/2, n_1 = \frac{32\log(6MH/\delta)}{\epsilon^2}$ and $n_2 = \frac{8\log(3MH/\delta)}{\epsilon}$.
**while** true **do**
    Fix a regressor $f \in \mathcal{F}$.
    Collect $n_1$ trajectories according to $\pi_f$ and estimate $V(\pi_f)$ via a Monte-Carlo estimate $\hat{V}(\pi_f)$.
    If $|\hat{V}(\pi_f) - \hat{V}^\star| \le \epsilon_{\text{demand}}$, return $\pi_f$.
    Otherwise update $\mathcal{F}$ by calling DFS-LEARN $(p, \mathcal{F}, \epsilon, \delta/(3MH^2n_2))$ on each of the $H-1$
prefixes $p$ of each of the first $n_2$ paths collected for the Monte-Carlo estimate.
**end while**

---

*trajectories.*

*Proof.* Since we ran DFS-LEARN at $\varnothing$, we may apply Theorem 5. By specification of the algorithm, we certainly ran TD-ELIM at $\varnothing$, which is at level $h = 1$, so we apply the conclusions in Theorem 3. In particular, we know that $f^\star \in \mathcal{F}$ and that for any surviving $f \in \mathcal{F}$,

$$|\hat{V}^f(p, \pi_f) - V^\star| = |\hat{V}^f(p, \pi_f) - V^f(p, \pi_f) + V^f(p, \pi_f) - V^{f^\star}(p, \pi_{f^\star})|$$

$$\le \frac{\phi}{\sqrt{12}} + 8\phi\sqrt{K} + 2\phi + 20(H-1)\sqrt{K}\phi \le \epsilon/8.$$

The last bound follows from the setting of $\phi$ and $\tau_1$. Since our estimate $\hat{V}^\star$ is $\hat{V}^f(p, \pi_f)$ for some surviving $f$, we guarantee estimation error at most $\epsilon/8$.

As for the sample complexity, Theorem 5 shows that the total number of executions of TD-ELIM can be at most $MH$, which is our setting of $T$. $\square$

Finally we turn to the EXPLORE-ON-DEMAND routine.

**Theorem 6** (Guarantee for EXPLORE-ON-DEMAND). *Consider running* EXPLORE-ON-DEMAND *with regressors $\mathcal{F}$, estimate $\hat{V}^\star$ and parameters $\epsilon, \delta$ and assume that $|\hat{V}^\star - V^\star| \le \epsilon/8$. Then with probability at least $1 - \delta$,* EXPLORE-ON-DEMAND *terminates after at most,*

$$\tilde{\mathcal{O}}\left(\frac{MH^6K^2}{\epsilon^3}\log(N/\delta)\log(1/\delta)\right)$$

*trajectories and it returns a policy $\pi_f$ with $V^\star - V(\varnothing, \pi_f) \le \epsilon$.*

See Appendix H for details.

## D  Proof of Theorem 1

The proof of the main theorem follows from straightforward application of Theorems 5 and 6. First, since we run DFS-LEARN at the root, $\varnothing$, the bias and estimation bounds in Theorem 3 apply at $\varnothing$, so we guarantee accurate estimation of the value $V^\star$ (See Corollary 1). This is required by the EXPLORE-ON-DEMAND routine, but at this point, we can simply apply Theorem 6, which is guaranteed to find a $\epsilon$-suboptimal policy and also terminate in $MH$ iterations. Combining these two results, appropriately allocating the failure probability $\delta$ evenly across the two calls, and accumulating the sample complexity bounds establishes Theorem 1.

## E  Proof of Theorem 3

The proof of Theorem 3 is quite technical, and we compartmentalize into several components. Throughout we will use the preconditions of the theorem, which we reproduce here.

**Condition 1.** For all $f \in \mathcal{F}$ and $a \in \mathcal{A}$, we have estimates $\hat{V}^f(p \circ a, \pi_f)$ such that,

$$|\hat{V}^f(p \circ a, \pi_f) - V^f(p \circ a, \pi_f)| \le \phi.$$

**Condition 2.** For all $f, g \in \mathcal{F}$ and $a \in \mathcal{A}$ we have,

$$|V^f(p \circ a, \pi_f) - V^g(p \circ a, \pi_g)| \leq \tau_1.$$

We will make frequent use of the parameters $\phi$ and $\tau_1$ which are specified by these two conditions, and explicit in the theorem statement.

Recall the notation,

$$V^f(p, \pi_g) \triangleq \mathbb{E}_{x \sim D_p} f(x, \pi_g(x)),$$

which will be used heavily throughout the proof.

We will suppress dependence on the distribution $D_p$, since we are considering one invocation of TD-ELIM and we always roll into $p$. This means that all (observation, reward) tuples will be drawn from $D_p$. Secondly it will be convenient to introduce the shorthand $V^f(p) = V^f(p, \pi_f)$ and similarly for the estimates. Finally, we will further shorten the value functions for paths $p \circ a$ by defining,

$$V_a^f \triangleq \mathbb{E}_{x \sim D_{p \circ a}} f(x, \pi_f(x)) = V^f(p \circ a, \pi_f).$$

We will also use $\hat{V}_a^f$ to denote the estimated versions which we have according to Condition 1.

Lastly, our proof makes extensive use of the following random variable, which is defined for a particular regressor $f \in \mathcal{F}$:

$$Y(f) \triangleq (f(x, a) - r(a) - \hat{V}^f(p \circ a))^2 - (f^\star(x, a) - r(a) - \hat{V}^{f^\star}(p \circ a))^2.$$

Here $(x, r) \sim D_p$ and $a \in \mathcal{A}$ is drawn uniformly at random as prescribed by Algorithm 7. We use $Y(f)$ to denote the random variable associated with regressor $f$, but sometimes drop the dependence on $f$ when it is clear from context.

To proceed, we first compute the expectation and variance of this random variable.

**Lemma 1** (Properties of TD Squared Loss). *Assume Condition 1 holds. Then for any $f \in \mathcal{F}$, the random variable $Y$ satisfies,*

$$\mathbb{E}_{x,a,r}[Y] = \mathbb{E}_{x,a}\left[(f(x,a) - \hat{V}^f(p \circ a) - f^\star(x,a) + V^{f^\star}(p \circ a))^2\right] - \mathbb{E}_{x,a}\left[(\hat{V}^{f^\star}(p \circ a) - V^{f^\star}(p \circ a))^2\right]$$

$$\underset{x,a,r}{\mathrm{Var}}[Y] \leq 32\mathbb{E}_{x,a}[Y] + 64\phi^2.$$

*Proof.* For shorthand, denote $f = f(x,a)$, $f^\star = f^\star(x,a)$ and recall the definition of $V_a^f$ and $\hat{V}_a^f$.

$$\mathbb{E}_{x,a,r} Y$$
$$= \mathbb{E}_{x,a,r}\left[(f - \hat{V}_a^f - r(a))^2 - (f^\star - \hat{V}_a^{f^\star} - r(a))^2\right]$$
$$= \mathbb{E}_{x,a,r}\left[(f - \hat{V}_a^f)^2 - 2r(a)(f - \hat{V}_a^f - f^\star + \hat{V}_a^{f^\star}) - (f^\star - \hat{V}_a^{f^\star})^2\right]$$

Now recall that $\mathbb{E}[r(a)|x,a] = f^*(x,a) - V_a^{f^\star}$ by definition of $f^*$, which allows us to deduce,

$$\mathbb{E}_{x,a,r} Y$$
$$= \mathbb{E}_{x,a}\left[(f - \hat{V}_a^f)^2 - 2(f^\star - V_a^{f^\star})(f - \hat{V}_a^f) + 2(f^\star - \hat{V}_a^{f^\star} + \hat{V}_a^{f^\star} - V_a^{f^\star})(f^\star - \hat{V}_a^{f^\star}) - (f^\star - \hat{V}_a^{f^\star})^2\right]$$
$$= \mathbb{E}_{x,a}\left[(f - \hat{V}_a^f)^2 - 2(f^\star - V_a^{f^\star})(f - \hat{V}_a^f) + (f^\star - \hat{V}_a^{f^\star})^2 + 2(\hat{V}_a^{f^\star} - V_a^{f^\star})(f^\star - \hat{V}_a^{f^\star})\right]$$
$$= \mathbb{E}_{x,a}\left[(f - \hat{V}_a^f)^2 - 2(f^\star - V_a^{f^\star})(f - \hat{V}_a^f) + (f^\star - V_a^{f^\star} + V_a^{f^\star} - \hat{V}_a^{f^\star})^2 + 2(\hat{V}_a^{f^\star} - V_a^{f^\star})(f^\star - \hat{V}_a^{f^\star})\right]$$
$$= \mathbb{E}_{x,a}\left[(f - \hat{V}_a^f - f^\star + V_a^{f^\star})^2 + 2(V_a^{f^\star} - \hat{V}_a^{f^\star})(f^\star - V_a^{f^\star}) + (V_a^{f^\star} - \hat{V}_a^{f^\star})^2 + 2(\hat{V}_a^{f^\star} - V_a^{f^\star})(f^\star - \hat{V}_a^{f^\star})\right]$$
$$= \mathbb{E}_{x,a}\left[(f - \hat{V}_a^f - f^\star + V_a^{f^\star})^2 - (V_a^{f^\star} - \hat{V}_a^{f^\star})^2\right].$$

For the second claim, notice that we can write,

$$Y = (f - \hat{V}_a^f - f^\star + \hat{V}_a^{f^\star})(f - \hat{V}_a^f + f^\star - \hat{V}_a^{f^\star} - 2r(a)),$$

so that,
$$Y^2 \leq 16(f - \hat{V}_a^f - f^\star + \hat{V}_a^{f^\star})^2.$$
This holds because all quantities in the second term are bounded in $[0, 1]$. Therefore,

$$
\begin{aligned}
\mathrm{Var}(Y) &\leq \mathbb{E}[Y^2] \\
&\leq 16\mathbb{E}_{x,a}\left[(f(x,a) - \hat{V}_a^f - f^\star(x,a) + \hat{V}_a^{f^\star})^2\right] \\
&= 16\mathbb{E}_{x,a}\left[(f(x,a) - \hat{V}_a^f - f^\star(x,a) + V_a^{f^\star} + \hat{V}_a^{f^\star} - V_a^{f^\star})^2\right] \\
&\leq 32\mathbb{E}_{x,a}\left[(f(x,a) - \hat{V}_a^f - f^\star(x,a) + V_a^{f^\star})^2\right] + 32\phi^2 \\
&\leq 32\mathbb{E}_{x,a}Y + 64\phi^2
\end{aligned}
$$

The first inequality is straightforward, while the second inequality is from the argument above. The third inequality uses the fact that $(a + b)^2 \leq 2a^2 + 2b^2$ and the fact that for each $a$, the estimate $\hat{V}_a^{f^\star}$ has absolute error at most $\phi$ (By Condition 1). The last inequality adds and subtracts the term involving $(V_a^{f^\star} - \hat{V}_a^{f^\star})^2$ to obtain $\mathbb{E}_{x,a}Y$. $\qquad \square$

The next step is to relate the empirical squared loss to the population squared loss, which is done by application of Bernstein's inequality.

**Lemma 2** (Squared Loss Deviation Bounds). *Assume Condition 1 holds. With probability at least $1 - \delta/2$, where $\delta$ is a parameter of the algorithm, $f^\star$ survives the filtering step of Algorithm 7 and moreover, any surviving $f$ satisfies,*

$$\mathbb{E}Y(f) \leq 6\phi^2 + \frac{120\log(2N/\delta)}{n_{train}}.$$

*Proof.* We will apply Bernstein's inequality on the centered random variable,

$$\sum_{i=1}^{n_{\text{train}}} Y_i(f) - \mathbb{E}Y_i(f),$$

and then take a union bound over all $f \in \mathcal{F}$. Here the expectation is over the $n_{\text{train}}$ samples $(x_i, a_i, r_i)$ where $(x_i, r) \sim D_p$, $a_i$ is chosen uniformly at random, and $r_i = r(a_i)$. Notice that since actions are chosen uniformly at random, all terms in the sum are identically distributed, so that $\mathbb{E}Y_i(f) = \mathbb{E}Y(f)$.

To that end, fix one $f \in \mathcal{F}$ and notice that $|Y - \mathbb{E}Y| \leq 8$ almost surely, as each quantity in the definition of $Y$ is bounded in $[0, 1]$, so each of the four terms can be at most $4$, but two are non-positive and two are non-negative in $Y - \mathbb{E}Y$. We will use Lemma 1 to control the variance. Bernstein's inequality implies that, with probability at least $1 - \delta$,

$$
\begin{aligned}
\sum_{i=1}^{n_{\text{train}}} \mathbb{E}Y_i - Y_i &\leq \sqrt{2\sum_i \mathrm{Var}(Y_i)\log(1/\delta)} + \frac{16\log(1/\delta)}{3} \\
&\leq \sqrt{64\sum_i(\mathbb{E}(Y_i) + 2\phi^2)\log(1/\delta)} + \frac{16\log(1/\delta)}{3}
\end{aligned}
$$

The first inequality here is Bernstein's inequality while the second is based on the variance bound in Lemma 1.

Now letting $X = \sqrt{\sum_i(\mathbb{E}(Y_i) + 2\phi^2)}$, $Z = \sum_i Y_i$ and $C = \sqrt{\log(1/\delta)}$, the inequality above is equivalent to,

$$
\begin{aligned}
X^2 - 2n_{\text{train}}\phi^2 - Z &\leq 8XC + \frac{16}{3}C^2 \\
\Rightarrow X^2 - 8XC + 16C^2 - Z &\leq 2n_{\text{train}}\phi^2 + 22C^2 \\
\Rightarrow (X - 4C)^2 - Z &\leq 2n_{\text{train}}\phi^2 + 22C^2 \\
\Rightarrow -Z &\leq 2n_{\text{train}}\phi^2 + 22C^2.
\end{aligned}
$$

Using the definition of $-Z$, this last inequality implies

$$\sum_{i=1}^{n_{\text{train}}}(f^{\star}(x_i, a_i) - r_i(a_i) - \hat{V}^{f^{\star}}(p \circ a_i))^2 \leq \sum_{i=1}^{n_{\text{train}}}(f(x_i, a_i) - r_i(a_i) - \hat{V}^{f}(p \circ a_i))^2 + 2n_{\text{train}}\phi^2 + 22\log(1/\delta).$$

Via a union bound over all $f \in \mathcal{F}$, rebinding $\delta \leftarrow \delta/(2N)$, and dividing through by $n_{\text{train}}$, we have,

$$\tilde{R}(f^{\star}) \leq \min_{f \in \mathcal{F}}\tilde{R}(f) + 2\phi^2 + \frac{22\log(2N/\delta)}{n_{\text{train}}}.$$

Since this is precisely the threshold used in filtering regressors, we ensure that $f^{\star}$ survives.

Now for any surviving regressor $f$, we are ensured that $Z$ is upper bounded in the elimination step (6). Specifically we have,

$$(X - 4C)^2 \leq Z + 2n_{\text{train}}\phi^2 + 22C^2 \leq 4n_{\text{train}}\phi^2 + 44C^2$$
$$\Rightarrow X^2 \leq (\sqrt{4n_{\text{train}}\phi^2 + 44C^2} + 4C)^2$$
$$\leq 8n_{\text{train}}\phi^2 + 120C^2.$$

This proves the claim since $X^2 = n_{\text{train}}\mathbb{E}Y(f) + 2n_{\text{train}}\phi^2$ (Recall that the $Y_i$s are identically distributed). $\qquad\square$

This deviation bound allows us to establish the three claims in Theorem 3. We start with the estimation error claim, which is straightforward.

**Lemma 3** (Estimation Error). *Let $\delta \in (0, 1)$. Then with probability at least $1 - \delta$, for all $f \in \mathcal{F}$ that are retained by the Algorithm 7, we have estimates $\hat{V}^f(p, \pi_f)$ with,*

$$|\hat{V}^f(p, \pi_f) - V^f(p, \pi_f)| \leq \sqrt{\frac{2\log(2N/\delta)}{n_{train}}}.$$

*Proof.* The proof is a consequence of Hoeffding's inequality and a union bound. Clearly the Monte Carlo estimate,

$$\hat{V}^f(p, \pi_f) = \frac{1}{n_{\text{train}}}\sum_{i=1}^{n_{\text{train}}}f(x_i, \pi_f(x_i)),$$

is unbiased for $V^f(p, \pi_f)$ and the centered quantity is bounded in $[-1, 1]$. Thus Hoeffding's inequality gives precisely the bound in the lemma. $\qquad\square$

Next we turn to the claim regarding bias.

**Lemma 4** (Bias Accumulation). *Assume Conditions 1 and 2 hold. In the same $1 - \delta/2$ event in Lemma 2, for any pair $f, g \in \mathcal{F}$ retained by Algorithm 7, we have,*

$$V^f(p, \pi_f) - V^g(p, \pi_g) \leq 2\sqrt{K}\sqrt{7\phi^2 + \frac{120\log(2N/\delta)}{n_{train}}} + 2\phi + \tau_1$$

*Proof.* Throughout the proof, we use $\mathbb{E}_x[\cdot]$ to denote expectation when $x \sim D_p$. We start by expanding definitions,

$$V^f(p, \pi_f) - V^g(p, \pi_g) = \mathbb{E}_x[f(x, \pi_f(x)) - g(x, \pi_g(x))]$$

Now, since $g$ prefers $\pi_g(x)$ to $\pi_f(x)$, it must be the case that $g(x, \pi_g(x)) \geq g(x, \pi_f(x))$, so that,

$$V^f(p, \pi_f) - V^g(p, \pi_g) \leq \quad \mathbb{E}_x f(x, \pi_f(x)) - g(x, \pi_f(x))$$
$$= \quad \mathbb{E}_x[f(x, \pi_f(x)) - \hat{V}^f(p \circ \pi_f(x), \pi_f) - f^{\star}(x, \pi_f(x)) + V^{f^{\star}}(p \circ \pi_f(x), \pi_{f^{\star}})]$$
$$- \mathbb{E}_x[g(x, \pi_f(x)) - \hat{V}^g(p \circ \pi_f(x), \pi_g) - f^{\star}(x, \pi_f(x)) + V^{f^{\star}}(p \circ \pi_f(x), \pi_{f^{\star}})]$$
$$+ \mathbb{E}_x[\hat{V}^f(p \circ \pi_f(x), \pi_f) - \hat{V}^g(p \circ \pi_f(x), \pi_g)].$$

This last equality is just based on adding and subtracting terms. The first two terms look similar, and we will relate them to the squared loss. For the first, by Lemma 1, we have that for each $x \in \mathcal{X}$,

$$\mathbb{E}_{r,a|x}[Y(f)] + \mathbb{E}_{a|x}[(\hat{V}^{f^\star}(p \circ a, \pi_{f^\star}) - V^{f^\star}(p \circ a, \pi_{f^\star}))^2]$$

$$= \mathbb{E}_{a|x}\left[(f(x,a) - \hat{V}^f(p \circ a, \pi_f) - f^\star(x,a) + V^{f^\star}(p \circ a, \pi_{f^\star}))^2\right]$$

$$\geq \frac{1}{K}\left[(f(x,\pi_f(x)) - \hat{V}^f(p \circ \pi_f(x), \pi_f) - f^\star(x,\pi_f(x)) + V^{f^\star}(p \circ \pi_f(x), \pi_{f^\star}))^2\right].$$

The equality is Lemma 1 while the inequality follows from the fact that each action, in particular $\pi_f(x)$, is played with probability $1/K$ and the quantity inside the expectation is non-negative. Now by Jensen's inequality the first term can be upper bounded as,

$$\mathbb{E}_x[f(x,\pi_f(x)) - \hat{V}^f(p \circ \pi_f(x), \pi_f) - f^\star(x,\pi_f(x)) + V^{f^\star}(p \circ \pi_f(x), \pi_{f^\star})]$$

$$\leq \sqrt{\mathbb{E}_x[(f(x,\pi_f(x)) - \hat{V}^f(p \circ \pi_f(x), \pi_f) - f^\star(x,\pi_f(x)) + V^{f^\star}(p \circ \pi_f(x), \pi_{f^\star}))^2]}$$

$$= \sqrt{K\mathbb{E}_x\left[\frac{1}{K}(f(x,\pi_f(x)) - \hat{V}^f(p \circ \pi_f(x), \pi_f) - f^\star(x,\pi_f(x)) + V^{f^\star}(p \circ \pi_f(x), \pi_{f^\star}))^2\right]}$$

$$\leq \sqrt{K\left(\mathbb{E}_{x,a,r}[Y(f)] + \mathbb{E}_{x,a}[(\hat{V}^{f^\star}(p \circ a, \pi_{f^\star}) - V^{f^\star}(p \circ a, \pi_{f^\star}))^2]\right)}$$

$$\leq \sqrt{K}\sqrt{\mathbb{E}Y(f) + \phi^2}$$

$$\leq \sqrt{K}\sqrt{7\phi^2 + \frac{120\log(N/\delta)}{n_{\text{train}}}},$$

where the last step follows from Lemma 2. This bounds the first term in the expansion of $V^f(p, \pi_f) - V^g(p, \pi_g)$. Now for the term involving $g$, we can apply essentially the same argument,

$$-\mathbb{E}_x[g(x,\pi_f(x)) - \hat{V}^g(p \circ \pi_f(x), \pi_g) - f^\star(x,\pi_f(x)) + V^{f^\star}(p \circ \pi_f(x), \pi_{f^\star})]$$

$$\leq \sqrt{\mathbb{E}_x[(g(x,\pi_f(x)) - \hat{V}^g(p \circ \pi_f(x), \pi_g) - f^\star(x,\pi_f(x)) + V^{f^\star}(p \circ \pi_f(x), \pi_{f^\star}))^2]}$$

$$\leq \sqrt{K}\sqrt{7\phi^2 + \frac{120\log(N/\delta)}{n_{\text{train}}}}$$

Summarizing, the current bound we have is,

$$V^f(p, \pi_f) - V^g(p, \pi_g) \leq 2\sqrt{K}\sqrt{7\phi^2 + \frac{120\log(N/\delta)}{n_{\text{train}}}} + \mathbb{E}_x[\hat{V}^f(p \circ \pi_f(x), \pi_f) - \hat{V}^g(p \circ \pi_f(x), \pi_g)]$$

(10)

The last term is easily bounded by the preconditions in Theorem 3. For each $a$, we have,

$\hat{V}^f(p \circ a, \pi_f) - \hat{V}^g(p \circ a, \pi_g)$

$\leq |\hat{V}^f(p \circ a, \pi_f) - V^f(p \circ a, \pi_f)| + |V^f(p \circ a, \pi_f) - V^g(p \circ a, \pi_g)| + |V^g(p \circ a, \pi_g) - \hat{V}^g(p \circ a, \pi_g)|$

$\leq 2\phi + \tau_1,$

from Conditions 1 and 2. Consequently,

$$\mathbb{E}_x[\hat{V}^f(p \circ \pi_f(x), \pi_f) - \hat{V}^g(p \circ \pi_f(x), \pi_g)]$$

$$= \sum_{a \in \mathcal{A}} \mathbb{E}_x\left[\mathbf{1}[\pi_f(x) = a](\hat{V}^f(p \circ a, \pi_f) - \hat{V}^g(p \circ a, \pi_g))\right]$$

$$\leq 2\phi + \tau_1.$$

This proves the claim. $\qquad\square$

Lastly, we must show how the squared loss relates to the risk, which helps establish the last claim of the theorem. The proof is similar to that of the bias bound but has subtle differences that require reproducing the argument.

**Lemma 5** (Instantaneous Risk Bound). *Assume Conditions 1 and 2 hold. In the same $1 - \delta/2$ event in Lemma 2, for any regressor $f \in \mathcal{F}$ retained by Algorithm 7, we have,*

$$V^{f^\star}(p, \pi_{f^\star}) - V^{f^\star}(p, \pi_f) \leq \sqrt{2K}\sqrt{7\phi^2 + \frac{120\log(2N/\delta)}{n_{train}}} + 2(\phi + \tau_1).$$

*Proof.*

$$V^{f^\star}(p, \pi_{f^\star}) - V^{f^\star}(p, \pi_f) = \mathbb{E}_x[f^\star(x, \pi_{f^\star}(x)) - f^\star(x, \pi_f(x))]$$
$$\leq \mathbb{E}_x[f^\star(x, \pi_{f^\star}(x)) - f(x, \pi_{f^\star}(x)) + f(x, \pi_f(x)) - f^\star(x, \pi_f(x))].$$

This follows since $f$ prefers its own action to that of $f^\star$, so that $f(x, \pi_f(x)) \geq f(x, \pi_{f^\star}(x))$. For any observation $x \in \mathcal{X}$ and action $a \in \mathcal{A}$, define,

$$\Delta_{x,a} = (f(x, a) - \hat{V}^f(p \circ a) - f^\star(x, a) + V^{f^\star}(p \circ a)),$$

where $V^f(p) = \mathbb{E}_{x \sim D_p}[f(x, \pi_f(x))]$ and similarly for $\hat{V}^p()$. Then we can write,

$$V^{f^\star}(p, \pi_{f^\star}) - V^{f^\star}(p, \pi_f)$$
$$\leq \mathbb{E}_x[\Delta_{x,\pi_f(x)} - \Delta_{x,\pi_{f^\star}(x)} + \hat{V}^f(p \circ \pi_f(x)) - V^{f^\star}(p \circ \pi_f(x)) - \hat{V}^f(p \circ \pi_{f^\star}(x)) + V^{f^\star}(p \circ \pi_{f^\star}(x))].$$

The term involving both $\Delta$s can be bounded as in the proof of Lemma 4. For any $x \in \mathcal{X}$

$$\mathbb{E}_{r,a|x}Y(f) + \mathbb{E}_{a|x}[(\hat{V}^{f^\star}(p \circ a) - V^{f^\star}(p \circ a))^2]$$
$$= \mathbb{E}_{a|x}\left[(f(x, a) - \hat{V}^f(p \circ a) - f^\star(x, a) + V^{f^\star}(p \circ a))^2\right]$$
$$\geq \frac{\Delta^2_{x,\pi_f(x)} + \Delta^2_{x,\pi_{f^\star}(x)}}{K} \geq \frac{(\Delta_{x,\pi_{f^\star}(x)} - \Delta_{x,\pi_f(x)})^2}{2K}.$$

Thus,

$$\mathbb{E}_x[\Delta_{x,\pi_f(x)} - \Delta_{x,\pi_{f^\star}(x)}] \leq \sqrt{2K\mathbb{E}\frac{(\Delta_{x,\pi_f(x)} - \Delta_{x,\pi_{f^\star}(x)})^2}{2K}}$$

$$\leq \sqrt{2K}\sqrt{\mathbb{E}Y(f) + \phi^2} \leq \sqrt{2K}\sqrt{7\phi^2 + \frac{120\log(2N/\delta)}{n_{\text{train}}}}.$$

We are left to bound the residual term,

$$(\hat{V}^f(p \circ \pi_f(x)) - V^{f^\star}(p \circ \pi_f(x)) - \hat{V}^f(p \circ \pi_{f^\star}(x)) + V^{f^\star}(p \circ \pi_{f^\star}(x)))$$
$$\leq \left|V^f(p \circ \pi_f(x)) - V^{f^\star}(p \circ \pi_f(x)) - V^f(p \circ \pi_{f^\star}(x)) + V^{f^\star}(p \circ \pi_{f^\star}(x))\right| + 2\phi$$
$$\leq 2(\phi + \tau_1).$$

$\square$

Notice that Lemma 5 above controls the quantity $V^{f^\star}(p, \pi_{f^\star}) - V^{f^\star}(p, \pi_f)$ which is the difference in values of the optimal behavior from $p$ and the policy that first acts according to $\pi_f$ and then behaves optimally thereafter. This is *not* the same as acting according to $\pi_f$ for all subsequent actions. We will control this cumulative risk $V^\star(p) - V(p, \pi_f)$ in the second phase of the algorithm.

**Proof of Theorem 3:** Equipped with the above lemmas, we can proceed to prove the theorem. By assumption of the theorem, Conditions 1 and 2 hold, so all lemmas are applicable. Apply Lemma 3 with failure probability $\delta/2$, where $\delta$ is the parameter in the algorithm, and apply Lemma 2, which also fails with probability at most $\delta/2$. A union bound over these two events implies that the failure probability of the algorithm is at most $\delta$.

Outside of this failure event, all three of Lemmas 3, 4, and 5 hold. If we set $n_{\text{train}} = 24\log(4N/\delta)/\phi^2$ then these four bounds give,

$$|\hat{V}^f(p, \pi_f) - V^f(p, \pi_f)| \leq \frac{\phi}{\sqrt{12}}$$
$$|V^f(p, \pi_f) - V^g(p, \pi_g)| \leq 8\phi\sqrt{K} + 2\phi + \tau_1$$
$$V^{f^\star}(p, \pi_{f^\star}) - V^{f^\star}(p, \pi_f) \leq 4\phi\sqrt{2K} + 2\phi + 2\tau_1.$$

These bounds hold for all $f, g \in \mathcal{F}$ that are retained by the algorithm. Of course by Lemma 2, we are also ensured that $f^{\star}$ is retained by the algorithm.

# F   Proof of Theorem 4

This result is a straightforward application of Hoeffding's inequality. We collect $n_{\text{test}}$ observations $x_i \sim D_p$ by applying path $p$ from the root and use the Monte Carlo estimates,

$$\hat{V}^f(p, \pi_f) = \frac{1}{n_{\text{test}}} \sum_{i=1}^{n_{\text{test}}} f(x_i, \pi_f(x_i)).$$

By Hoeffding's inequality, via a union bound over all $f \in \mathcal{F}$, we have that with probability at least $1 - \delta$,

$$\left| \hat{V}^f(p, \pi_f) - V^f(p, \pi_f) \right| \leq \sqrt{\frac{2 \log(2N/\delta)}{n_{\text{test}}}}.$$

Setting $n_{\text{test}} = 2 \log(2N/\delta)/\phi^2$, gives that our empirical estimates are at most $\phi$ away from the population versions.

Now for the first claim, if the population versions are already within $\tau_2$ of each other, then the empirical versions are at most $2\phi + \tau_2$ apart by the triangle inequality,

$$|\hat{V}^f(p, \pi_f) - \hat{V}^g(p, \pi_g)| \leq |\hat{V}^f(p, \pi_f) - V^f(p, \pi_f)| + |V^f(p, \pi_f) - V^g(p, \pi_g)| + |V^g(p, \pi_g) - \hat{V}^g(p, \pi_g)|$$
$$\leq 2\phi + \tau_2.$$

This applies for any pair $f, g \in \mathcal{F}$ whose population value predictions are within $\tau_2$ of each other. Since we set $\epsilon_{\text{test}} \geq 2\phi + \tau_2$ in Theorem 4, this implies that the procedure returns $\texttt{true}$.

For the second claim, if the procedure returns $\texttt{true}$, then all empirical value predictions are at most $\epsilon_{\text{test}}$ apart, so the population versions are at most $2\phi + \epsilon_{\text{test}}$ apart, again by the triangle inequality. Specifically, for any pair $f, g \in \mathcal{F}$ we have,

$$|V^f(p, \pi_f) - V^g(p, \pi_g)| \leq |V^f(p, \pi_f) - \hat{V}^f(p, \pi_f)| + |\hat{V}^f(p, \pi_f) - \hat{V}^g(p, \pi_g)| + |\hat{V}^g(p, \pi_g) - V^g(p, \pi_g)|$$
$$\leq 2\phi + \epsilon_{\text{test}}.$$

Both arguments apply for all pairs $f, g \in \mathcal{F}$, which proves the claim.

# G   Proof of Theorem 5

Assume that all calls to TD-ELIM and CONSENSUS operate successfully, i.e., we can apply Theorems 3 and 4 on any path $p$ for which the appropriate subroutine has been invoked. We will bound the number of calls and hence the total failure probability.

Recall that $\epsilon$ is the error parameter passed to DFS-LEARN and that we set $\phi = \frac{\epsilon}{320 H^2 \sqrt{K}}$.

We first argue that in all calls to TD-ELIM, the estimation precondition is satisfied. To see this, notice that by design, the algorithm only calls TD-ELIM at path $p$ after the recursive step, which means that for each $a$, we either ran TD-ELIM on $p \circ a$ or CONSENSUS returned $\texttt{true}$ on $p \circ a$. Since both Theorems 3 and 4 guarantee estimation error of order $\phi$, the estimation precondition for path $p$ holds. This argument applies to all paths $p$ for which we call TD-ELIM, so that the estimation precondition is always satisfied.

We next analyze the bias term, for which proceed by induction. To state the inductive claim, we define the notion of an *accessed path*. We say that a path $p$ is *accessed* if either (a) we called TD-ELIM on path $p$ or (b) we called CONSENSUS on $p$ and it returned $\texttt{true}$.

The induction is on the number of actions remaining, which we denote with $\eta$. At time point $h$ there are $H - h + 1$ actions remaining.

**Inductive Claim:** For all accessed paths $p$ with $\eta$ actions remaining and any pair $f, g \in \mathcal{F}$ of surviving regressors,

$$|V^f(p, \pi_f) - V^g(p, \pi_g)| \leq 20\eta\sqrt{K}\phi.$$

**Base Case:** The claim clearly holds when $\eta = 0$ since there are zero actions remaining and all regressors estimate future reward as zero.

**Inductive Step:** Assume that the inductive claim holds for all accessed paths with $\eta - 1$ actions remaining. Consider any accessed path $p$ with $\eta$ actions remaining. Since we access the path $p$, either we call TD-ELIM or CONSENSUS returns `true`. If we call TD-ELIM, then we access the paths $p \circ a$ for all $a \in \mathcal{A}$. By the inductive hypothesis, we have already filtered the regressor class so that for all $a \in \mathcal{A}, f, g \in \mathcal{F}$, we have,

$$|V^f(p \circ a, \pi_f) - V^g(p \circ a, \pi_f)| \leq 20(\eta - 1)\sqrt{K}\phi.$$

We instantiate $\tau_1 = 20(\eta - 1)\sqrt{K}\phi$ in the bias precondition of Theorem 3. We also know that the estimation precondition is satisfied with parameter $\phi$. The bias bound of Theorem 3 shows that, for all $f, g \in \mathcal{F}$ retained by the algorithm,

$$|V^f(p, \pi_f) - V^g(p, \pi_g)| \leq 8\phi\sqrt{K} + 2\phi + \tau_1$$
$$\leq 10\phi\sqrt{K} + 20(\eta - 1)\phi\sqrt{K} \leq 20(\eta - \frac{1}{2})\phi\sqrt{K}. \qquad (11)$$

Thus, the inductive step holds in this case.

The other case we must consider is if CONSENSUS returns `true`. Notice that for a path $p$ with $\eta$ actions to go, we call CONSENSUS with parameter $\epsilon_{\text{test}} = 20(\eta - 1/4)\sqrt{K}\phi$. We actually invoke the routine on path $p$ when we are currently processing a path $p'$ with $\eta + 1$ actions to go (i.e., $p = p' \circ a$ for some $a \in \mathcal{A}$), so we set $\epsilon_{\text{test}}$ in terms of $H - |p'| - 5/4 = \eta - 1/4$. ($|p|$ is actually one less than the level of the state reached by applying $p$ from the root.) Then, by Theorem 4, we have the bias bound,

$$|V^f(p, \pi_f) - V^g(p, \pi_f)| \leq 2\phi + 20(\eta - 1/4)\sqrt{K}\phi$$
$$\leq 20\eta\sqrt{K}\phi.$$

Thus, we have established the inductive claim.

**Verifying preconditions for Theorem 3:** To apply the conclusions of Theorem 3 at some state $s$, we must verify that the preconditions hold, with the appropriate parameter settings, before we execute TD-ELIM. We saw above that the estimation precondition always holds with parameter $\phi$, assuming successful execution of all subroutines. The inductive argument also shows that the bias precondition also holds with $\tau_1 = 20(\eta - 1)\sqrt{K}\phi$ for a state $s \in \mathcal{S}_{H-\eta+1}$ that we called TD-ELIM on. Thus, both preconditions are satisfied at each execution of TD-ELIM, so the conclusions of Theorem 3 apply at any state $s$ for which we have executed the subroutine. Note that the precondition parameters that we use here, specifically $\tau_1$, depend on the actions-to-go $\eta$.

Substituting the level $h$ for the actions-to-go $\eta$ gives $\tau_1 = 20(H - h)\sqrt{K}\phi$ at level $h$.

**Sample Complexity:** We now bound the number of calls to each subroutine, which reveals how to allocate the failure probability and gives the sample complexity bound. Again assume that all calls succeed.

First notice that if we call CONSENSUS on some state $s$ with $\eta$ actions-to-go for which we have already called TD-ELIM, then CONSENSUS returns `true` (assuming all calls to subroutines succeed). This follows because TD-ELIM guarantees that the population predicted values are at most $20(\eta - 1/2)\sqrt{K}\phi$ apart (Eq. (11)), which becomes the choice of $\tau_2$ in application of Theorem 4. This is valid since,

$$2\phi + 20(\eta - 1/2)\sqrt{K}\phi \leq 20(\eta - 1/4)\sqrt{K}\phi = \epsilon_{\text{test}},$$

so that the precondition for Theorem 4 holds. Thus, at any level $h$, we can call TD-ELIM at most one time per state $s \in \mathcal{S}_h$. In total, this yields $MH$ calls to TD-ELIM.

Next, since we only make recursive calls when we execute TD-ELIM, we expand at most $M$ paths per level. This means that we call CONSENSUS on at most $MK$ paths per level, since the fan-out of the tree is $K$. Thus, the number of calls to CONSENSUS is at most $MKH$.

By our setting $\delta$ in the subroutine calls (i.e. $\delta/(2MKH)$ in calls to CONSENSUS and $\delta/(2MH)$ in calls to TD-ELIM), and by Theorems 3 and 4, the total failure probability is therefore at most $\delta$.

Each execution of TD-ELIM requires $n_{\text{train}}$ trajectories while executions of CONSENSUS require $n_{\text{test}}$ trajectories. Since before each execution of TD-ELIM we always perform $K$ executions of CONSENSUS, if we perform $T$ executions of TD-ELIM, the total sample complexity is bounded by,

$$T(n_{\text{train}} + Kn_{\text{test}}) \leq (3 \times 10^6)\frac{TH^4K}{\epsilon^2}\log(8NMH/\delta) + (3 \times 10^5)\frac{TH^4K^2}{\epsilon^2}\log(4NMKH/\delta)$$
$$= \mathcal{O}\left(\frac{TH^4K^2}{\epsilon^2}\log\left(\frac{NMHK}{\delta}\right)\right).$$

The total number of executions of TD-ELIM can be no more than $MH$, by the argument above.

# H Analysis for EXPLORE-ON-DEMAND

Throughout the proof, assume that $|\hat{V}^\star - V^\star| \leq \epsilon/8$. We will ensure that the first half of the algorithm guarantees this. Let $\mathcal{E}$ denote the event that all Monte-Carlo estimates $\hat{V}(\varnothing, \pi_f)$ are accurate and all calls to DFS-LEARN succeed (so that we may apply Theorem 5). By accurate, we mean,

$$|\hat{V}(\varnothing, \pi_f) - V(\varnothing, \pi_f)| \leq \epsilon/8.$$

Formally, $\mathcal{E}$ is the intersection over all executions of DFS-LEARN of the event that the conclusions of Theorem 5 apply for this execution and the intersection over all iterations of the loop in EXPLORE-ON-DEMAND of the event that the Monte Carlo estimate $\hat{V}(\varnothing, \pi_f)$ is within $\epsilon/8$ of $V(\varnothing, \pi_f)$. We will bound this failure probability, i.e. $\mathbb{P}[\bar{\mathcal{E}}]$, toward the end of the proof.

**Lemma 6** (Risk bound upon termination). *If $\mathcal{E}$ holds, then when EXPLORE-ON-DEMAND terminates, it outputs a policy $\pi_f$ with $V^\star - V(\pi_f) \leq \epsilon$.*

*Proof.* The proof is straightforward.

$$V^\star - V(\pi_f) \leq |V^\star - \hat{V}^\star| + |\hat{V}^\star - \hat{V}(\pi_f)| + |\hat{V}(\pi_f) - V(\pi_f)|$$
$$\leq \epsilon/8 + \epsilon/2 + \epsilon/8 = 3\epsilon/4 \leq \epsilon.$$

The first bound follows by assumption on $\hat{V}^\star$ while the second comes from the definition of $\epsilon_{\text{demand}}$ and the third holds under event $\mathcal{E}$. □

**Lemma 7** (Termination Guarantee). *If $\mathcal{E}$ holds, then when EXPLORE-ON-DEMAND selects a policy that is at most $\epsilon/4$-suboptimal, it terminates.*

*Proof.* We must show that the test succeeds, for which we will apply the triangle inequality,

$$|\hat{V}^\star - \hat{V}(\pi_f)| \leq |\hat{V}^\star - V^\star| + |V^\star - V(\pi_f)| + |V(\pi_f) - \hat{V}(\pi_f)|$$
$$\leq \epsilon/8 + \epsilon/4 + \epsilon/8 \leq \epsilon/2 = \epsilon_{\text{demand}}.$$

Therefore the test is guaranteed to succeed. Again the last bound here holds under event $\mathcal{E}$. □

At some point in the execution of the algorithm, define a set of *learned states* $L$ as

$$L(\mathcal{F}) \triangleq \bigcup_h \left\{s \in \mathcal{S}_h : \max_{f \in \mathcal{F}} V^\star(s) - V^{f^\star}(s, \pi_f) \leq 4\phi\sqrt{2K} + 2\phi + 40(H - h)\sqrt{K}\phi\right\}. \quad (12)$$

By Theorem 3, any state for which we have successfully called TD-ELIM is $L(\mathcal{F})$, since the condition is precisely the instantaneous risk bound. Since we only ever call TD-ELIM through DFS-LEARN, the fact that these calls to TD-ELIM succeeded is implied by the event $\mathcal{E}$. The *unlearned states* are denoted $\bar{L}$, where the dependence on $\mathcal{F}$ is left implicit.

For a policy $\pi_f$, let $q^{\pi_f}[s \to \bar{L}]$ denote the probability that when behaving according to $\pi_f$ starting from state $s$, we visit an unlearned state. We now show that $q^{\pi_f}[\varnothing \to \bar{L}]$ is related to the risk of the policy $\pi_f$.

**Lemma 8** (Policy Risk). *Define $L$ as in Eq. (12) and define $q^{\pi_f}[s \to \bar{L}]$ accordingly. Assume that $\mathcal{E}$ holds and let $f$ be a surviving regressor, so that $\pi_f$ is a surviving policy. Then,*

$$V^\star - V(\varnothing, \pi_f) \leq q^{\pi_f}[\varnothing \to \bar{L}] + 40\sqrt{K}\phi H^2.$$

*Proof.* Recall that under event $\mathcal{E}$, we can apply the conclusions of Theorem 3 with $\phi = \frac{\epsilon}{320H^2\sqrt{K}}$ and $\tau_1 = 20(H-h)\sqrt{K}\phi$ for any $h$ and state $s \in \mathcal{S}_h$ for which we have called TD-ELIM. Our proof proceeds by creating a recurrence relation through application of Theorem 3 and then solving the relation. Specifically, we want to prove the following inductive claim.

**Inductive Claim:** For a state $s \in L$ with $\eta$ actions to go,

$$V^\star(s) - V(s, \pi_f) \leq 40\phi\sqrt{K}\eta^2 + q^{\pi_f}[s \to \bar{L}].$$

**Base Case:** With zero actions to go, all policies achieve zero reward and no policies visit $\bar{L}$ from this point, so the inductive claim trivially holds.

**Inductive Step:** For the inductive hypothesis, consider some state $s$ at level $h$, for which TD-ELIM has successfully been called. There are $\eta = H - h + 1$ actions to go. By Theorem 5, we know that,

$$V^\star(s) - V^{f^\star}(s, \pi_f) \leq 4\phi\sqrt{2K} + 2\phi + 2\tau_1,$$

with $\tau_1 = 20(H-h)\phi\sqrt{K}$. This bound is clearly at most $40\eta\phi\sqrt{K}$. Now,

$$V^\star(s) - V(s, \pi_f) = V^\star(s) - V^{f^\star}(s, \pi_f) + V^{f^\star}(s, \pi_f) - V(s, \pi_f)$$
$$\leq 40\eta\phi\sqrt{K} + \mathbb{E}_{(x,r)\sim D_s} r(\pi_f(x)) + V^\star(s \circ \pi_f(x)) - r(\pi_f(x)) - V(s \circ \pi_f(x), \pi_f).$$

Let us focus on just the second term, which is equal to,

$$\mathbb{E}_{x\sim D_s}\left[(V^\star(s \circ \pi_f(x)) - V(s \circ \pi_f(x), \pi_f))\left(\mathbf{1}[\Gamma(s, \pi_f(x)) \in L] + \mathbf{1}[\Gamma(s, \pi_f(x)) \notin L]\right)\right]$$
$$\leq \sum_{s' \in L} \mathbb{P}_{x\sim D_s}[\Gamma(s, \pi_f(x)) = s']\left(V^\star(s') - V(s', \pi_f)\right) + \mathbb{P}_{x\sim D_s}[\Gamma(s, \pi_f(x)) \notin L].$$

Since all of the recursive terms above correspond only to states $s' \in L$, we may apply the inductive hypothesis, to obtain the bound,

$$40\eta\phi\sqrt{K} + \sum_{s' \in L} \mathbb{P}_{x \in D_s}[\Gamma(s, \pi_f(x)) = s']\left(40(h-1)^2\phi\sqrt{K} + q^{\pi_f}[s' \to \bar{L}]\right) + \mathbb{P}_{x\sim D_s}[\Gamma(s, \pi_f(x)) \notin L]$$
$$\leq 40\eta\phi\sqrt{K} + 40(\eta-1)^2\phi\sqrt{K} + q^{\pi_f}[s \to \bar{L}]$$
$$\leq 40\phi\sqrt{K}\eta^2 + q^{\pi_f}[s \to \bar{L}].$$

Thus, we have proved the inductive claim. Applying at the root of the tree gives the result. $\square$

Recall that we set $\phi = \frac{\epsilon}{320H^2\sqrt{K}}$ in DFS-LEARN. This ensures that $40H^2\phi\sqrt{K} \leq \epsilon/8$, which means that if $q^{\pi_f}[\varnothing \to \bar{L}] = 0$, then we ensure $V^\star - V(\varnothing, \pi_f) \leq \epsilon/8$.

**Lemma 9** (Each non-terminal iteration makes progress). *Assume that $\mathcal{E}$ holds. If $\pi_f$ is selected but fails the test, then with probability at least $1 - \exp(-\epsilon n_2/8)$, at least one of the $n_2$ trajectories collected visits a state $s \notin L$.*

*Proof.* First, if $\pi_f$ fails the test, we know that,

$$\epsilon_{\text{demand}} < |\hat{V}(\varnothing, \pi_f) - \hat{V}^\star| \leq \epsilon/4 + |V(\varnothing, \pi_f) - V^\star|,$$

which implies that,

$$\epsilon/4 < V^\star - V(\varnothing, \pi_f).$$

On the other hand Lemma 8, shows that,

$$V^\star - V(\varnothing, \pi_f) \leq q^{\pi_f}[\varnothing \to \bar{L}] + 40H^2\sqrt{K}\phi.$$

Using our setting of $\phi$, and combining the two bounds gives,

$$\epsilon/4 < q^{\pi_f}[\varnothing \to \bar{L}] + \epsilon/8 \Rightarrow q^{\pi_f}[\varnothing \to \bar{L}] > \epsilon/8.$$

Thus, the probability that all $n_2$ trajectories miss $\bar{L}$ is,

$$\mathbb{P}[\text{all trajectories miss } \bar{L}] = (1 - q^{\pi_f}[\varnothing \to \bar{L}])^{n_2}$$
$$\leq (1 - \epsilon/8)^{n_2} \leq \exp(-\epsilon n_2/8).$$

Therefore, we must hit $\bar{L}$ with substantial probability. $\square$

## H.1 Proof of Theorem 6

Again assume that $\mathcal{E}$ holds. First, by Lemma 6, we argued that if EXPLORE-ON-DEMAND terminates, then it outputs a policy that satisfies the PAC-guarantee. Moreover, by Lemma 7, we also argued that if EXPLORE-ON-DEMAND selects a policy that is at most $\epsilon/4$ suboptimal, then it terminates. Thus the goal of the proof is to show that it quickly finds a policy that is at most $\epsilon/4$ suboptimal.

Every execution of the loop in EXPLORE-ON-DEMAND either passes the test or fails the test at level $\epsilon_{\text{demand}}$. If the test succeeds, then Lemma 6 certifies that we have found an $\epsilon$-suboptimal policy, thus establishing the PAC-guarantee. If the test fails, then Lemma 9 guarantees that we call DFS-LEARN on a state that was not previously trained on. Thus at each non-terminal iteration of the loop, we call DFS-LEARN and hence TD-ELIM on at least one state $s \notin L$, so that the set of learned states grows by at least one. By Lemma 8 and our setting of $\phi$, if we have called TD-ELIM on all states at all levels, then we guarantee that all surviving policies have risk at most $\epsilon/8$. Thus the number of iterations of the loop is at most $MH$ since that is the number of unique states in the model.

**Bounding** $\mathbb{P}[\bar{\mathcal{E}}]$**:** Since we have bounded the total number of iterations, we are now in a position to assign failure probabilities and bound the event $\mathcal{E}$. Actually we must consider not only the event $\mathcal{E}$ but also the event that all non-terminal iterations visit some state $s \notin L$. Call this new event $\mathcal{E}'$ which is the intersection of $\mathcal{E}$ with the event that all unsuccessful iterations visit $\bar{L}$.

More formally, we use the fact that for events $A_0, \ldots, A_t$, we have,

$$\mathbb{P}[\bigcup_{i=0}^{t} A_i] \leq \mathbb{P}[A_0] + \sum_{i=1}^{t} \mathbb{P}[A_i|\bar{A}_0, \ldots, \bar{A}_{i-1}]. \tag{13}$$

This inequality is based on applying the union bound to the events $A_i' = (A_i \cap \bigcap_{j=0}^{i-1} A_j)$.

Our analysis above bounds events of this form, namely the probability of a failure event conditioned on no previous failure event occurring. Specifically, we decompose $\mathcal{E}'$ into three types of events.

1. $B_t^{(1)}$ denotes the event that the Monte Carlo estimate $\hat{V}(\varnothing, \pi_f)$ is accurate for the $t^{\text{th}}$ iteration of the while loop.

2. $B_t^{(2)}$ denotes the event that DFS-LEARN succeeds at the $t^{\text{th}}$ iteration of the while loop.

3. $B_t^{(3)}$ denotes the event that $t$ is a non-terminal iteration and we visit $\bar{L}$ at the $t^{\text{th}}$ iteration.

These events are defined for $t \in [MH]$, since we know that if all events hold we will perform at most $MH$ iterations. $\mathcal{E}'$ is the intersection of all of these events.

The failure probability can be expressed as,

$$\mathbb{P}[\bar{\mathcal{E}}'] = \mathbb{P}[\bigcup_{t=1}^{MH} \bar{B}_t^{(1)} \cup \bar{B}_t^{(2)} \cup \bar{B}_t^{(3)}],$$

and via Equation 13, it suffices to bound each event, conditioned on all previous success events.

We have $\delta$ probability to allocate, and since we perform at most $MH$ iterations, we allocate $\delta/(MH)$ probability to each iteration and $1/3$ of the available failure probability to each type of event.

For the initial Monte-Carlo estimate in event $B_t^{(1)}$, by Hoeffding's inequality, we know that,

$$|\hat{V}(\varnothing, \pi_f) - V(\varnothing, \pi_f)| \leq \sqrt{\frac{\log(6MH/\delta)}{2n_1}}.$$

We want this bound to be at most $\epsilon/8$ which requires,

$$n_1 \geq \frac{32 \log(6MH/\delta)}{\epsilon^2}.$$

This bound holds for any fixed $\pi_f$, and it is independent of previous events.

For the second event, for each of the $Hn_2$ calls to DFS-LEARN, we set the parameter to be $\delta/(3MH^2n_2)$, so that by Theorem 5, we may apply Theorem 3 at all states that we have called TD-ELIM on. Again this bounds the probability of $\bar{B}_t^{(2)}$, independently of previous events.

Finally, conditioned on $B_t^{(1)}$, we may apply Lemma 9 at iteration $t$ to observe that the the conditional probability of $\bar{B}_t^{(3)}$ is at most $\exp(-n_2\epsilon/8)$. And for this to be smaller than $\delta/(3MH)$ we require,

$$n_2 \geq \frac{8\log(3MH/\delta)}{\epsilon}.$$

Both conditions on $n_1$ and $n_2$ are met by our choices in the algorithm specification.

In total, if we set, $n_1 = \frac{32\log(6MH/\delta)}{\epsilon^2}$ and $n_2 = 8\log(3MH/\delta)/\epsilon$ in EXPLORE-ON-DEMAND and if EXPLORE-ON-DEMAND always call DFS-LEARN with parameter $\delta/(3MH^2n_2)$ we guarantee that the total failure probability for this subroutine is at most $\delta$.

**Sample Complexity:** It remains to bound the sample complexity for the execution of EXPLORE-ON-DEMAND. We do at most $MH$ iterations, and in each iteration we use $n_1$ trajectories to compute Monte-Carlo estimates, contributing an $MHn_1$ to the sample complexity. We also call DFS-LEARN on each of the $Hn_2$ prefixes collected during each iteration so that there are at most $MH^2n_2$ calls to DFS-LEARN in total. Naïvely, each call to DFS-LEARN takes at most $O(\frac{MH^5K^2}{\epsilon^2}\log(n_2NMKH/\delta))$ episodes, leading to a crude sample complexity bound of,

$$\tilde{\mathcal{O}}\left(\frac{M^2H^7K^2}{\epsilon^3}\log(N/\delta)\log(1/\delta)\right).$$

Recall that the $\tilde{\mathcal{O}}$ notation suppresses all logarithmic factors except those involving $N$ and $\delta$.

This bound can be significantly improved using a more careful argument. Apart from the first call to TD-ELIM in each application of DFS-LEARN, the total number of additional calls to TD-ELIM is bounded by $MH$ since once we call TD-ELIM on a state, CONSENSUS always returns true.

Each call to TD-ELIM requires $n_{\text{train}} + Kn_{\text{test}}$ samples (because we always call CONSENSUS on all direct descendants before), and the total number of calls is at most,

$$MH^2n_2 + MH = \mathcal{O}\left(\frac{MH^2}{\epsilon}\log(MH/\delta)\right).$$

With our settings of $n_{\text{train}}$ and $n_{\text{test}}$, the sample complexity is therefore at most,

$$\mathcal{O}\left(\frac{MH^6K^2}{\epsilon^3}\log(MHKN/(\epsilon\delta))\log(MH/\delta)\right)$$
$$= \tilde{\mathcal{O}}\left(\frac{MH^6K^2}{\epsilon^3}\log(N/\delta)\log(1/\delta)\right).$$

This concludes the proof of Theorem 6.