[Reviews · NeurIPS 2016]

Reviewer 1

Summary

This is a technical paper which proves PAC learnability of a complex model, but it does not make a convincing case that the model is a useful extension of standard MDPs.

Qualitative Assessment

The paper analyzes PAC learnability of finite-horizon Contextual MDPs. Contextual MDPs are a specific type of POMDPs with the restriction that the optimal q-function depends only on the most recent observation (instead of the belief state). The authors show that Contextual MDPs are not poly PAC learneable even when either memoryless policies are considered or value function approximation is used. However, when both memoryless policies and value function approximation is used and the transitions are deterministic, then the model is PAC learnable in a polynomial number of episodes (and the complexity is independent of the number of observations). The paper is well written overall. The proofs are quite clear and quite thorough. I am not quite sure that the 16 pages of technical proofs in the appendix are suitable for a conference; the paper may better fit a journal format. The presentation could be further improved by better describing the results and how they were obtained earlier on in the paper. It takes a lot of effort to really understand the contribution and what are the main ideas in constructing the new algorithm. The main limitation of the paper is that its results apply to a rather limited setting. While it is clear that Contextual MDPs are slightly more general than plain MDPs, the authors do not make a convincing case that any interesting or useful problems fall into this category. In particular the results apply to a POMDP in which 1) transitions are deterministic, 2) optimal q-function depends only on the most recent observation, and 3) the value function approximator can exactly represent the optimal q-function. The authors only spend a single paragraph on the justification of why this may be a useful model in Example 1, but the example is quite construed. There are very few (if any at all) plausible types of POMDPs in which the optimal q-value depends only on the last observation and action that are not also MDPs. I could imagine a model in which the optimal policy is reactive (memoryless), but the requirement of the value function to be memoryless is just too strong. The technical result of the paper is interesting in that it proposes a sampling method that is independent of the number of observations. It is a generalization of learnability results for MDP (with the caveat that the transitions are deterministic), but the proof and algorithms are much less elegant. Perhaps is the paper was posed in terms of learnability of MDPs with value function approximation, the results would be more meaningful and applicable. More detailed comments: 1) Introduction: the first few paragraphs are quite vague, the references (e.g. [21]) do not really support the claims, and the paper does not at all answer the question posed. 2) The layered process is really just a standard finite-horizon MDP (with non-stationary transitions) 3) Appendix A: Please start with the description of what is in the appendix. One needs to read the entire Theorem 2 to understand that it is just a technical lemma needed to prove Prop 1 & 2. 4) Section 2.2, Definition 1: This is a bit repetitive given the description in 2.1 5) Assumption 1: The requirement to have only deterministic transitions limits the applicability of the results quite a bit 6) Section 3.1: The algorithms and basic ideas of the proof are described very nicely in this section. 7) References have several instances of incorrect capitalization 8) Line 404: The argument regarding no need to consider stochastic policies is not convincing. I guess the equivalence follows from “splitting policies in MDPs”, but it would be good to provide more details.

Confidence in this Review

2-Confident (read it all; understood it all reasonably well)


Reviewer 2

Summary

The paper provides a reinforcement learning algorithm for a certain class of episodic MDPs, with PAC-style polynomial bounds on the sample complexity of the algorithm. In this so-called contextual-MDP model, while only a random output signal is observed rather than the state, it is assumed that the optimal Q-value function (hence the optimal action) is independent of the state given this signal. A finite class of predictors (measurement-dependent Q-functions) is given, one of which is assumed to be the true optimal one (realizability assumption), so that learning can proceed by eliminating predictors that are not consistent with Bellman's optimality equation. The model is further specialized to state deterministic transitions. The main result provides a sample-complexity bound which is polynomial in the model parameters, and in particular in the horizon size. Some lower bounds are developed to support the necessity of the main model assumptions (restriction to reactive policies, and realizability) to achieve such polynomial dependence on the horizon. As the authors claim, this is the first polynomial sample-complexity bound for RL with general function approximation.

Qualitative Assessment

The paper is well written, the context of the results well laid out, the proposed algorithm incorporates novel ideas, and the technical content is significant. Unfortunately, the model assumptions under which the main results are obtained are highly restrictive. The realizability assumption is clearly unrealistic with any reasonable class of finite value functions. This assumption is used in essential way in the proposed algorithm. The contextual MDP assumption also considerably limits the class of models included in the discussion. This assumption, together with the deterministic model assumption, eliminates any need for state estimation in the optimal policy. Overall, I am not convinced what lessons can be learned under such specialized assumptions. In my opinion this takes away from the contribution of the paper. Minor comments: 1. On page 2 middle: I am not sure why D_s belones to \Delta(…,[0,1]^K). Why the ^K? Isn't the last argument just the reward r (same for any action)? 2. Algorithm 1 line 2: The definition of \hat{V}^* is not clear, what does "for any f" mean here?

Confidence in this Review

2-Confident (read it all; understood it all reasonably well)


Reviewer 3

Summary

The paper formulates a new variant of MDP that incorporates contexts, here called Contextual-MDP. Having done so, the paper studies a number of possible assumptions for this problem and the consequences thereof. It considers both 1. restricting the policy class (for purposes of regret comparison) to reactive/memoryless policies, and 2. assuming the realizability of Q*, i.e. how well the true observation-action value function is approximated by some member of a given class of functions. In the form of a pair of lower bound results, the paper demonstrates that neither assumption is enough on its own: for each assumption, there exists a MDP that cannot be learned with a sub-exponential number of samples. However, the paper goes on to prove that both assumptions in conjunction enable PAC-learnability in a polynomial number of samples, and provide an algorithm (CTXMDPLEARN) for that purpose. Along the way, connections to contextual bandits and general MDPs are briefly highlighted.

Qualitative Assessment

Overall I found this paper very readable and almost immediately interesting, which is a nice and surprising thing for a paper outside my area of expertise. I was initially skeptical about the validity of the assumptions placed on MDPs, but the pair of lower-bound results did a good job of justifying them. Similarly, I appreciated not only the PAC-learnability result for Contextual-MDPs but the discussion surrounding it, which included a nice comparison to the two well-known models it generalizes (contextual bandits and MDPs). I can't say the algorithm (CTXMDPLEARN) is exactly intuitive, but its discussion in section 3.1 was well-written enough for me to get a rough picture of its workings, which is more than I can say for a lot of papers. My final interpretation: the results are interesting and connect to a number of topics in learning theory (contextual bandits and MDPs), and the model has a strong theoretical base and justification. This is why I scored a 4 for technical quality and novelty/originality. I only scored a 3 for potential impact because right now it's not immediately clear (at least to me) why a non-theoretical audience might care about these results; a few more sentences about applications might help. I scored a 4 for clarity and presentation because, while there are no figures and I would like more explanation of certain assertions (see Minor Feedback below), this is a well-written paper that thoughtfully and methodically argues for why the problem it poses is relevant and where the solution it provides sits in the context of existing work. For the reasons above I recommend this paper for acceptance, with my own non-expertise in the area being the primary reason against a stronger recommendation. Minor feedback: there were a few places where I wanted either a reference to literature or further explanation, for example: 1. Line 71, which asserts that "layered structure...is mathematically equivalent to an alternative reformulation without layered structure". That still doesn't seem obvious to me. 2. Line 195 says that a KWIK oracle "cannot exist even for simple problems." I'm not sure this is general enough knowledge to include uncited (it was at any rate news to me). 3. Line 309 says that previous polynomial sample complexity bounds for RL in infinite observation spaces require "prohibitively strong assumptions". It might be nice to have a reference to this previous work and a sentence or two about why those assumptions are prohibitively strong. 4. A few sentences about the applicability of this kind of work would be nice. There are a couple of mentions of utility for vision (line 26, line 151). A few more would probably help a non-theory audience, which I guess most of NIPS is.

Confidence in this Review

2-Confident (read it all; understood it all reasonably well)


Reviewer 4

Summary

The paper suggests the notion of Contextual MDPs for POMDPs satisfying certain properties. The authors suggest an algorithm for learning the true predictor from a given finite set with PAC guarantees.

Qualitative Assessment

The paper supplies several interesting results, but it has some flows which are hard for me to accept: 1. Semantics - The notion of Contextual MDPs was introduced before: Hallak, Assaf, Dotan Di Castro, and Shie Mannor. "Contextual Markov Decision Processes." arXiv preprint arXiv:1502.02259 (2015) In their definition, they assumed the "context" is unchanging - a more fitting setup for the name. In my view the setup described in the paper is much more related to POMDPs than to MDPs, so I found the authors' way of presentation odd. 2. The paper makes several very harsh assumptions: a. Shared observations => same Q's (by definition) --- In most real problems, same Q's => states equivalent (in terms of rewards and transition probabilities) which turns contextual MDPs to more of a state aggregation issue. This is also the example described in the paper. b. Deterministic transitions (Assumption 1). This assumption raises again the question of connection to MDPs - such an assumption should instead put the paper in the AI field. Needless to say, it also takes an already restricted model and restricts it further. I think the authors should have put more emphasis on why this assumption was needed and what can we do if it doesn't. c. Q* in F and F finite. Would be reasonable as the only assumption, but over the top with the rest of the assumptions. I find these assumptions too unreasonable and greatly hinders the impact and usefulness of the results. 3. Motivation - following my previous comments, I think some more emphasis on the motivation beyond such a model would have gone a great length in justifying the setup and assumptions. Update following rebuttal: My ratings have not changed. Perhaps the correct way to judge this work is from the "learning theory" perspective, but this is quite hard for me. I simply can't ignore the limited setup of the model and the lack of motivation beyond it. I advise the authors to either find a much more convincing motivating example, or to consider resubmitting under the "learning theory" umbrella topic.

Confidence in this Review

3-Expert (read the paper in detail, know the area, quite certain of my opinion)


Reviewer 5

Summary

The paper identifies an interesting yet useful special case of POMDP such that one can learn an approximate optimal policy in Poly(episode len, #actions, #states, log(#policies)) samples. There are three critical assumptions: (1) reactive policy: the policy only depends on the current observation (2) Q-function can be well approximated by a function class (several recent empirical successes of RL rely on this). (3) deterministic transitions. Even with the above 3 assumption, the problem is still quite challenging. In particular, the paper shows that assumption (1) or (2) alone is not enough for a result like this paper. The result of the paper seems to be a solid result. However, I find some statements of the paper can be misleading or imprecise, and the notations of the paper are very confusing to me, to a degree that I couldn't understand the technical content of the paper.

Qualitative Assessment

In fact, I don't understand the main technical content of the paper, due to the notations. I will list those parts which I found confusing. (1) sec 2.3., why the model reduces to stochastic contextual bandit? In a high level, it seems so. But I couldn't see the mapping immediately. I think it would be better to explain the details. (2) Assumption 1 appears in a rather strange place. Why not discuss it with other assumptions together? (3) page 2: is r the reward? does it only depend on the action? why use notation r(a)? (4) The paragraph about the predictability of Q^* is confusing: it is said that "for some POMDPs, we may be able to write Q^*....". Yes. But how restrictive is this should be discussed. Moreover, it would be better to define realizability more formally (you might want to say it is already defined. But one could ask why call it "realizable"? what is "not realizable"?). (5) there is no english description of the main algorithm. In fact, I can't understand it. Is \hat{V}^* a single value or a function (if a function, function of what)? I don't see what it is even after see eq(3). (6) It seems that V^* is the target optimal value. Do we need to update it? Otherwise, you already figure out the optimal value in the first step. (7) Now eq (3), what does x\sim D_{p'} mean?? p' is a path. Do you sample every node from the path, or just sample the lowest node of the path? (8) Right now, there are V^*, V, V^f and their hat versions. I have to guess and flip the paper back and forth to find their definitions (but still unclear to me what is what). (9) I think It would make sense to use English to state precisely and in details what the algorithm does. Currently, the paper spends a lot of space trying to convey the intuition. But one won't be able to appreciate the intuition if what the algorithm does is unclear.

Confidence in this Review

1-Less confident (might not have understood significant parts)


Reviewer 6

Summary

This paper introduces Contextual-MDPs, a model for reinforcement learning that is more general than regular MDPs, yet unlike POMDPs it may remain tractable. The paper also introduces a sample efficient but computationally inefficient algorithm to solve a restricted class of contextual-MDPs.

Qualitative Assessment

I am confused as to why example 1 cannot be described by a regular MDP. There exist multiple PAC-exploration algorithms that do not require a priori knowledge of the reachable parts of the state space [1,2,3,4]. The algorithms referenced above provide polynomial sample complexity bounds for reinforcement learning with infinite observation spaces. Each one of those algorithms uses a form of function approximation. Theorem 1 is neither the first polynomial sample complexity bound for reinforcement learning with infinite observation spaces, nor the first finite-sample guarantee for RL with function approximation (as claimed in lines 307-310). Additionally, none of those algorithms require deterministic transitions, nor do they require that Q^* is representable (line 200). Even for the restricted class of domains with deterministic transitions, assuming that Q^* is representable by a function approximator is a very strong assumption, far stronger than the assumptions made in those papers. In line 323, do you mean agrees with MDP results in the dependence on the policy complexity? Your bound does have a log(N) term. Overall, while I really like that the method presented in this paper is very different than existing PAC methods, I am not convinced that the algorithm presented offers any advantage compared to existing algorithms, or even that contextual-MDPs are necessary. [1] Pazis, Parr 2013. PAC optimal exploration in continuous space Markov decision processes [2] Grande, Walsh, How 2014, Sample efficient reinforcement learning with Gaussian processes [3] Pazis, Parr 2016, Efficient PAC-optimal Exploration in Concurrent, Continuous State MDPs with Delayed Updates [4] Liu, Guo, Brunskill 2016, PAC Continuous State Online Multitask Reinforcement Learning with Identification

Confidence in this Review

2-Confident (read it all; understood it all reasonably well)